# Identification of a herpes simplex virus 1 gene encoding neurovirulence factor by chemical proteomics

Akihisa Kato[1,2,3,8], Shungo Adachi [4,8], Shuichi Kawano [5], Kousuke Takeshima[1], Mizuki Watanabe[1], Shinobu Kitazume [6], Ryota Sato[7], Hideo Kusano[4], Naoto Koyanagi[1,2,3], Yuhei Maruzuru[1,2], Jun Arii[1,2,3], Tomohisa Hatta[4], Tohru Natsume [4✉] & Yasushi Kawaguchi [1,2,3✉]

Identification of the complete set of translated genes of viruses is important to understand viral replication and pathogenesis as well as for therapeutic approaches to control viral infection. Here, we use chemical proteomics, integrating bio-orthogonal non-canonical amino acid tagging and high-resolution mass spectrometry, to characterize the newly synthesized herpes simplex virus 1 (HSV-1) proteome in infected cells. In these infected cells, host cellular protein synthesis is shut-off, increasing the chance to preferentially detect viral proteomes. We identify nine previously cryptic orphan protein coding sequences whose translated products are expressed in HSV-1-infected cells. Functional characterization of one identified protein, designated piUL49, shows that it is critical for HSV-1 neurovirulence in vivo by regulating the activity of virally encoded dUTPase, a key enzyme that maintains accurate DNA replication. Our results demonstrate that cryptic orphan protein coding genes of HSV-1, and probably other large DNA viruses, remain to be identified.

[1] Division of Molecular Virology, Department of Microbiology and Immunology, The Institute of Medical Science, The University of Tokyo, Minato-ku, Tokyo 108-8639, Japan. [2] Department of Infectious Disease Control, International Research Center for Infectious Diseases, The Institute of Medical Science, The University of Tokyo, Minato-ku, Tokyo 108-8639, Japan. [3] Research Center for Asian Infectious Diseases, The Institute of Medical Science, The University of Tokyo, Minato-ku, Tokyo 108-8639, Japan. [4] Molecular Profiling Research Center for Drug Discovery (molprof), National Institute of Advanced Industrial Science and Technology (AIST), Tokyo 135-0064, Japan. [5] Department of Computer and Network Engineering, Graduate School of Informatics and Engineering, The University of Electro-Communications, Tokyo 182-8585, Japan. [6] Preparing Section for New Faculty of Medical Science, Fukushima Medical University, Fukushima City, Fukushima 960-1295, Japan. [7] Division of Innate Immunity, Department of Microbiology and Immunology, The Institute of Medical Science, The University of Tokyo, Minato-ku, Tokyo 108-8639, Japan. [8]These authors contributed equally: Akihisa Kato, Shungo Adachi. ✉email: t-natsume@aist.go.jp; ykawagu@ims.u-tokyo.ac.jp

Viruses package their genomes into virions for transmission to new hosts and this physical constraint tends to minimize their genome size. However, viruses need to produce many molecules from a limited set of viral genes to promote their replication, to hijack host cellular mechanisms and to evade host restriction mechanisms. To overcome this discrepancy, viruses have evolved multiple strategies: for example, viral genomes harbor a variety of non-canonical translational elements that allow them to densely pack coding information[1,2]. Notably, these elements usually lack well-defined sequence signatures, making it difficult to annotate translated sequences by traditional annotation approaches based on sequences. Therefore, new approaches that directly and comprehensively detect viral translational status are required to annotate yet-to-be unidentified viral translated sequences.

Herpesviruses are large DNA viruses that are ubiquitous pathogens in mammals, birds, reptiles, fishes, amphibians, and bivalves[3]. The genomes of these viruses were reported to encode approximately between 70 and 200 protein coding genes[3]. Recently, ribosome profiling, a powerful and comprehensive approach to map translated sequences in a genome[4], was used for a subset of herpesviruses. This technology clarified the high complexity of translations from the herpesvirus genomes and identified many previously unknown open reading frames (ORFs)[5–8]. However, although expression of the translated products from a fraction of the identified ORFs was confirmed in infected cells or in vivo[6], their biological significance in viral infection remains to be elucidated[8].

Herpes simplex virus 1 (HSV-1) is an extensively studied herpesvirus that causes a variety of human diseases including encephalitis, keratitis, and mucocutaneous and skin diseases such as herpes labialis, genital herpes, and herpetic whitlow[3]. To date, HSV-1 is thought to encode at least 84 different viral proteins, and unusually, only 2 HSV-1 proteins have introns in their coding domains[3]. To identify yet-to-be decoded protein coding genes of HSV-1, we employed the bio-orthogonal non-canonical amino acid tagging (BONCAT)-based mass spectrometry (MS) analyses[9], wherein a newly synthesized proteome in HSV-1-infected cells can be monitored. Because HSV-1 is known to shut-off host cellular protein synthesis[3,10,11], we assumed that this analysis has an advantage to preferentially identify translated HSV-1 gene products in infected cells compared with previously reported proteomic analyses of HSV-1-infected cells where selective identification of the HSV-1 proteome was profoundly hindered by the pre-existing cellular protein pool[8,12–14].

In this study, we identify nine previously cryptic orphan protein coding sequences (CDSs) whose translated products are expressed in HSV-1-infected cells. In-depth analyses of the translated product of one of the identified CDSs, designated piUL49, show that piUL49 is critical for HSV-1 neurovirulence in vivo by uniquely regulating the activity of virally encoded dUTPase, a key enzyme that maintains accurate DNA replication. Our results indicate that a complete understanding of viral pathogenesis requires not only their identification, but also their in-depth characterization.

## Results

### Decoding cryptic orphan HSV-1 CDSs by chemical proteomics.
In this study, we used mock-infected or wild-type HSV-1 F strain {HSV-1(F)} infected HeLa cells (incubated for 0.5–4 h or 8–12 h, respectively) labeled with azidohomoalanine (AHA), which functions as a surrogate for methionine[9]. Then, cells were harvested and lysed at 4 or 12 h post-infection (mock or HSV-1(F) groups, respectively; Fig. 1a). Newly synthesized proteins labeled with AHA in lysates of infected cells used for click reactions were affinity purified using alkyne-conjugating agarose resin, trypsinized on the resin, and subjected to tandem MS. In the experiments shown in Supplementary Fig 1, whereas the mean number of peptides detected in the presence of AHA labeling by MS was $2.73 \times 10^3$, that of peptides detected in the absence of AHA labeling was only 61.3, indicating the proper enrichment of newly synthesized proteins labeled with AHA by click reactions in these experiments.

At 4 h post-infection, the ion intensity of newly synthesized viral polypeptides was 37.5% of the ion intensity for total newly synthesized polypeptides (Fig. 1b). As expected, the frequency of newly synthesized HSV-1(F) polypeptides became dominant as viral infection progressed and increased to 85.3% at 12 h post-infection (Fig. 1b). Among peptides detected in HSV-1(F)-infected cells that were not mapped to host cellular ORFs in the experiments shown in Fig. 1a and Supplementary Fig. 1a, 97.2% mapped to reported HSV-1 protein coding genes, and 2.8% mapped to previously unidentified HSV-1 ORFs (Fig. 1c–e and Supplementary Data 1). Peptides from mock-infected cells did not mapped to any HSV-1 ORFs. Based on criteria that (i) at least two distinct peptides detected in HSV-1(F)-infected cells, each of which was identified by more than one peptide spectra match, were mapped to a previously cryptic orphan CDS, and (ii) the CDS was conserved in another HSV-1 strain {HSV-1 (17)}, we identified 10 previously cryptic orphan CDSs (Fig. 1f, Supplementary Fig. 2, and Supplementary Tables 1 and 2). Bioinformatic analyses using a sequence database of 80 HSV-1 strains revealed that 8 of the 10 identified CDSs were conserved in all HSV-1 strains, whereas other CDSs, the iUs1 and iUL50 genes, were conserved in 90.0 and 44.4% of HSV-1 strains, respectively (Supplementary Tables 2 and 3). Furthermore, transcripts encompassing each of the 10 identified CDSs were reported previously[8,15–19]. Of note, the recent publication by Whisnant et al., reported the decoding of HSV-1 by integrative functional genomics and the identification of ~180 potential cryptic ORFs, among which the expression of 12 ORFs in HSV-1-infected cells was verified at the peptide or protein level[8]. Among the 10 previously cryptic orphan CDSs identified in our study, only one CDS (iICP0, which was designated as RL2A in the publication) was also identified by Whisnant et al.[8].

### Expression of piUL49 in HSV-1-infected cells.
Among the identified previously cryptic orphan HSV-1 CDSs, we focused on one CDS, designated iUL49. Eight peptides detected in HSV-1(F)-infected cells were mapped to overall domains of iUL49 (Fig. 2a and Supplementary Fig. 2n). iUL49 potentially encodes 266 amino acids and lies within the HSV-1 UL49 gene, which encodes a virion tegument protein VP22 out of frame of the UL49 gene, indicating it encodes a previously unknown polypeptide (Fig. 2a and Supplementary Table 2). To confirm whether iUL49 expresses a previously unidentified viral protein (piUL49) in HSV-1-infected cells, we generated antisera raised against the carboxyl-terminal domain of piUL49 (Fig. 2a) and constructed a series of recombinant viruses: iUL49ΔM in which the predicted translational start codon of iUL49 was replaced by a threonine codon (iUL49ΔM); its repaired virus iUL49ΔM-rep; MEF-iUL49 and Venus-iUL49 in which iUL49 was tagged with an MEF tag consisting of Myc and Flag epitopes, and a Tobacco Etch Virus protease cleavage site or the fluorescent protein Venus, respectively; and MEF-iUL49ΔM and Venus-iUL49ΔM carrying the iUL49ΔM mutation in MEF-iUL49 and Venus-iUL49, respectively (Supplementary Fig. 3). As shown in Fig. 2b, c and Supplementary Fig. 4, immunoblotting with anti-GFP and anti-Flag antibodies detected a protein with an approximate molecular mass of 47 or 62 kDa from lysates of cells infected with Venus-

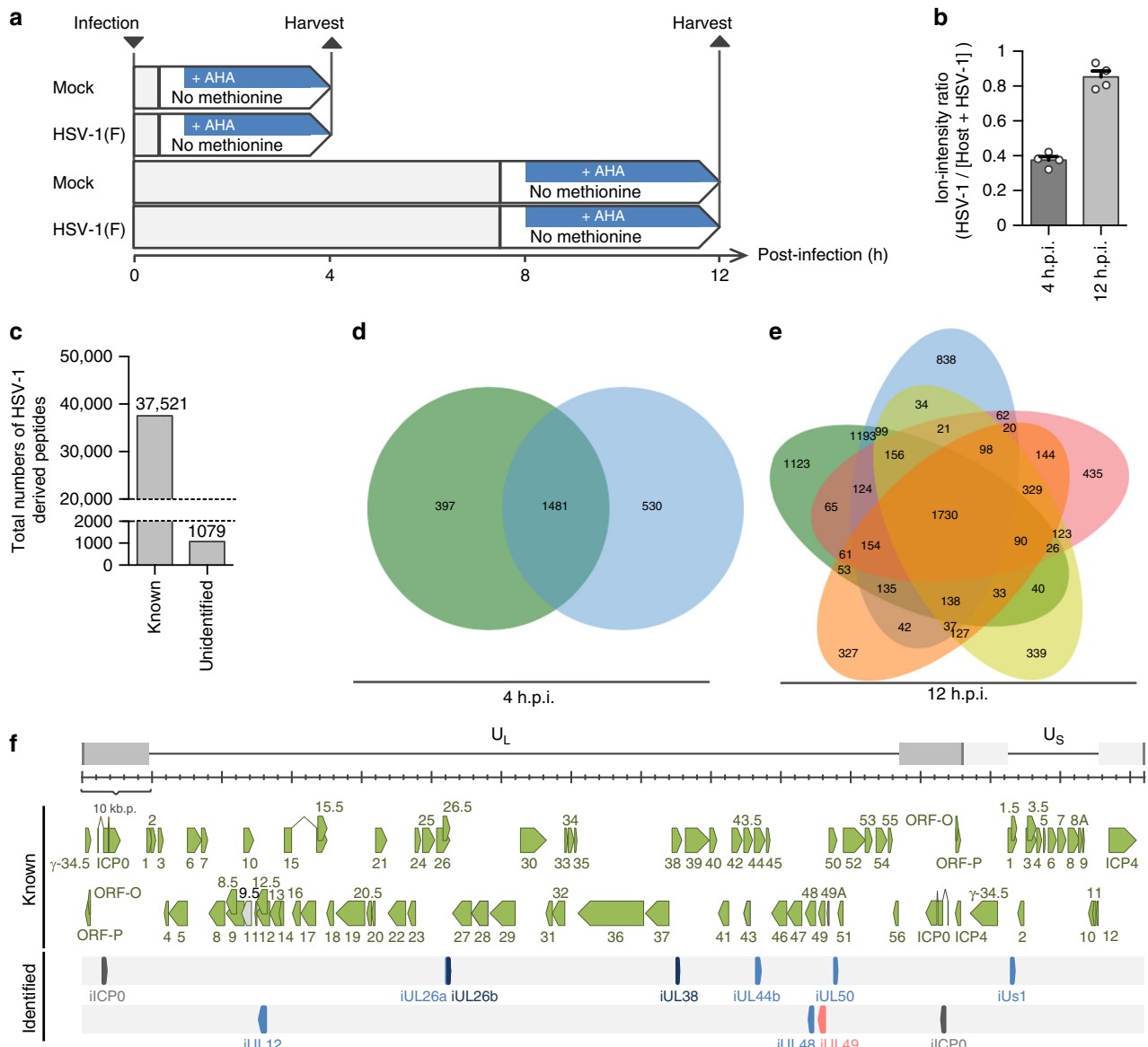

**Fig. 1 Identification of previously unidentified HSV-1 CDSs by B chemical proteomics. a** Diagram illustrating the procedure of AHA labeling experiments for BONCAT-based MS-analyses (n = 2). In each of the two experiments, MS-analyses were performed twice. **b** Ion-intensity of viral polypeptides relative to that of viral and host polypeptides. Each value is the mean ± SEM of four MS-analyses. **c** Total numbers of peptides detected in HSV-1-infected cells at 4 h and 12 h post-infection and mapped to known or previously unidentified HSV-1 CDSs in the experiments shown in Fig. 1a and Supplementary Fig. 1a. The number is shown above the bars. **d** Venn diagrams showing the results of two independent AHA-labeling experiments by 4 h post-infection using wild-type HSV-1(F)-infected cells as described in Fig. 1a. The numbers indicate the number of peptides derived from HSV-1 CDSs. **e** Venn diagrams showing the results of five independent AHA-labeling experiments by 12 h post-infection using wild-type HSV-1(F) infected cells as described in Fig. 1a (n = 2) and Supplementary Fig. 1a (n = 3). The numbers indicate the number of peptides derived from HSV-1 CDSs. **f** Schematic diagram of the genome of wild-type HSV-1 and the map of HSV-1 CDSs known and identified in this study. Source data are provided as a Source Data file.

iUL49 or MEF-iUL49, respectively, but not from cells mock-infected or infected with MEF-iUL49ΔM or Venus-iUL49ΔM. Similarly, antisera raised against piUL49 reacted with a protein with an approximate molecular mass of 33 kDa from lysates of cells infected with each of the various HSV-1 strains and iUL49ΔM-rep but not from cells mock-infected or infected with iUL49ΔM (Fig. 2d, e). The molecular masses of the MEF tag and Venus are approximately 12 and 29 kDa, respectively[20,21]. These results indicate that iUL49 is expressed in HSV-1-infected cells and encodes the previously unidentified protein (piUL49) with an approximate molecular mass of 33 kDa. It was reported that in the iUL49 gene locus, four major species of transcripts are expressed and each encompasses UL48, UL49, iUL49, and

UL49.5; UL48, UL49, and iUL49.5; UL49, iUL49, and UL49.5; or UL49 and iUL49 (Supplementary Fig. 5)[15], suggesting piUL49 is translated from a polycistronic transcript(s). In addition, we verified expression of another identified CDS, iUL44b, based on the observation that Vero cells infected with a recombinant virus Venus-iUL44b in which iUL44b was tagged with Venus (Supplementary Fig. 3) specifically produced a protein with an approximate molecular mass of 49 kDa (Supplementary Fig. 6).

**piUL49 is an HSV-1 virulence factor**. All VP22-null mutant viruses previously used to examine the effects of VP22 during HSV-1 infection harbor mutations that appear to also preclude

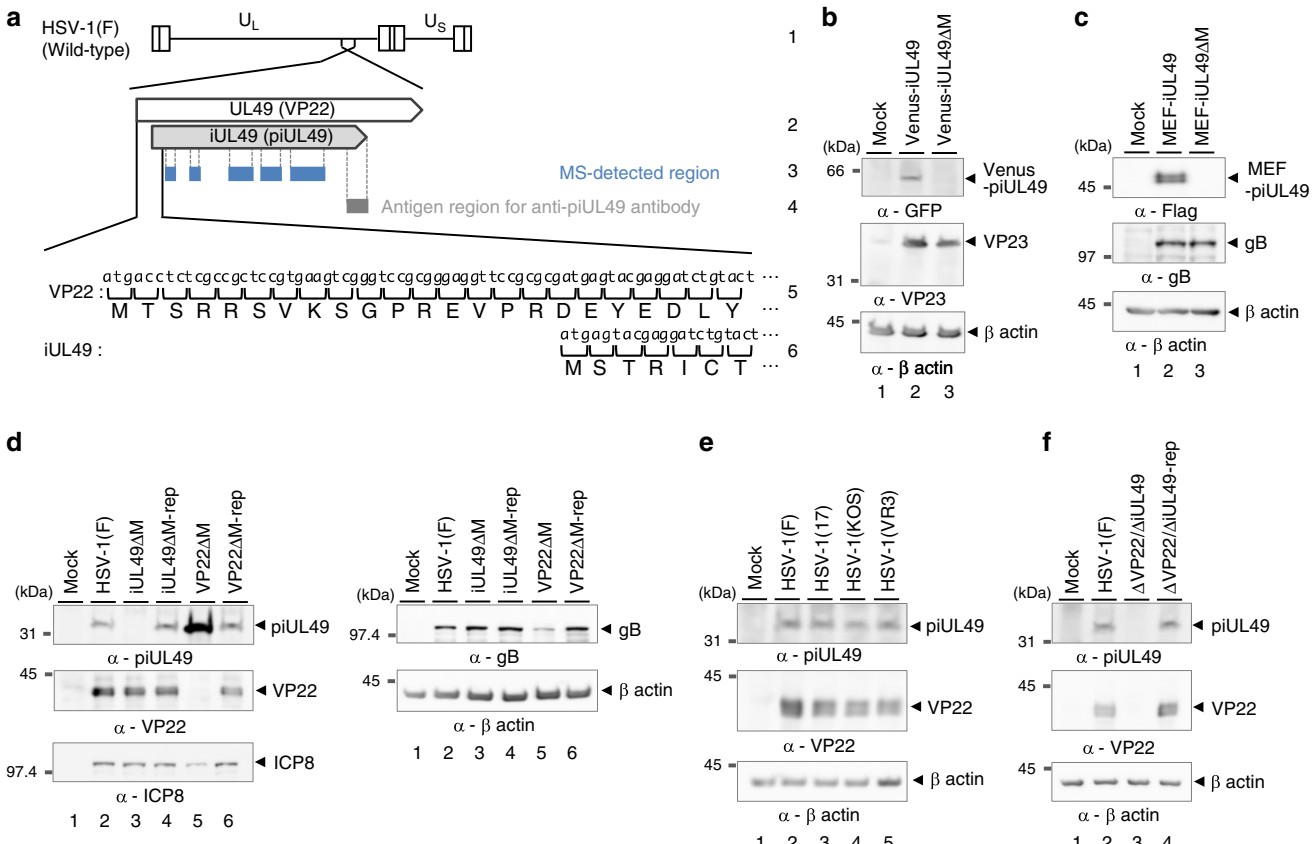

**Fig. 2 Expression of piUL49 in HSV-1-infected cells. a** Schematic diagram of the genome of wild-type HSV-1 (line 1) and the locations of the UL49 (VP22) and iUL49 (piUL49) CDSs (line 1). The domain(s) of iUL49 detected by MS-analyses (line 3) or used to generate antibodies to piUL49 (line 4). The nucleotide and amino-acid sequences of VP22 (line 5) and piUL49 (line 6) are indicated. **b, c** Vero cells mock-infected or infected with Venus-iUL49 (**b**), Venus-iUL49ΔM (**b**), MEF-iUL49 (**c**), or MEF-iUL49ΔM (**c**) for 12 h at an MOI of 10 were analyzed by immunoblotting with antibodies to GFP (**b**), VP23 (**b**), Flag (**c**), gB (**c**), or β actin (**b, c**). **d** Vero cells mock-infected or infected with wild-type HSV-1(F), iUL49ΔM, iUL49ΔM-rep, VP22ΔM, or VP22ΔM-rep for 12 h at an MOI of 10 were analyzed by immunoblotting with antibodies to piUL49, VP22, ICP8, gB, or β actin. **e, f** Vero cells mock-infected or infected with wild-type HSV-1(F) (**e, f**), HSV-1(17) (**e**), HSV-1(KOS) (**e**) or HSV-1(VR3) (**e**), ΔVP22/ΔiUL49 (**f**), or ΔVP22/ΔiUL49-rep (**f**) for 12 h at an MOI of 10 were analyzed by immunoblotting with antibodies to piUL49, VP22, or β actin. Digital images are representative of three independent experiments (**b**–**f**).

the expression of piUL49 (Supplementary Fig. 7). To dissect roles specific to VP22 or piUL49 in HSV-1-infected cells, we generated additional recombinant viruses: VP22ΔM in which the translational initiation codon of VP22 was substituted with a stop codon (VP22ΔM) and its repaired virus VP22ΔM-rep; ΔVP22/ΔiUL49 in which both the UL49 and iUL49 genes were disrupted by replacing VP22 codons 1–266, containing the entire coding sequence of iUL49, with a kanamycin resistance gene (ΔVP22/ΔiUL49) and its repaired virus ΔVP22/ΔiUL49-rep (Supplementary Fig. 8). As expected, cells infected with VP22ΔM or ΔVP22/ΔiUL49 did not produce VP22 or VP22 and piUL49, respectively (Fig. 2d, f). Notably, cells infected with iUL49ΔM accumulated VP22 at a level similar to cells infected with wild-type HSV-1(F) (Fig. 2d), whereas cells infected with VP22ΔM produced piUL49 at a level higher than cells infected with wild-type HSV-1(F) or VP22ΔM-rep (Fig. 2d). Similar observations were reported for gene products encoded by polycistronic transcripts[22,23], supporting our hypothesis that VP22 and piUL49 are translated from a polycistronic transcript(s).

Based on previous studies using VP22-null mutant viruses, VP22 was reported to be a positive regulator of HSV-1 replication in cell cultures and in vivo[24]. As reported previously[24], the VP22ΔM mutation significantly reduced progeny virus yields at multiplicities of infection (MOIs) of 10 and 0.01 (Fig. 3b, e) and plaque sizes (Fig. 3g, h) in Vero cells. In contrast, the iUL49ΔM

mutation had no effect on progeny virus yields or plaque sizes (Fig. 3a, d, g, h and Supplementary Fig. 9), and the ΔVP22/ΔiUL49 mutations reduced progeny virus yields and plaque sizes at levels similar to the VP22ΔM mutation (Fig. 3c, f–h). These results suggest that piUL49 has no obvious role in HSV-1 replication and cell–cell spread in these cells. In support of this conclusion, the iUL49ΔM mutation did not mis-localize VP16 or reduce the accumulation of ICP8 and gB in HSV-1-infected Vero cells in contrast to the VP22ΔM mutation (Fig. 2d and Supplementary Fig. 10a), as reported previously[24,25]. In addition, the iUL49ΔM mutation had no effect on the localization of VP22 (Supplementary Fig. 10b). Moreover, the VP22ΔM mutation augmented IL-1β secretion in HSV-1-infected J774A.1 cells in agreement with a previous report[26], whereas the iUL49ΔM mutation had no effect on IL-1β secretion (Supplementary Fig. 10c). This suggests that in contrast to VP22, piUL49 does not inhibit AIM2 inflammasome activation in these cells. We then tested the effects of piUL49 and/or VP22 on HSV-1 replication and virulence in mice following intracranial infection. As shown in Fig. 3i and l, the iUL49ΔM mutation augmented the 50% lethal dose (LD$_{50}$) 21.5-fold and reduced the progeny virus yield in brains 4.1-fold, indicating that piUL49 is critical for HSV-1 virulence and replication in vivo. Notably, the ΔVP22/ΔiUL49 mutations further augmented the LD$_{50}$ values and reduced progeny virus yields in brains compared with the iUL49ΔM

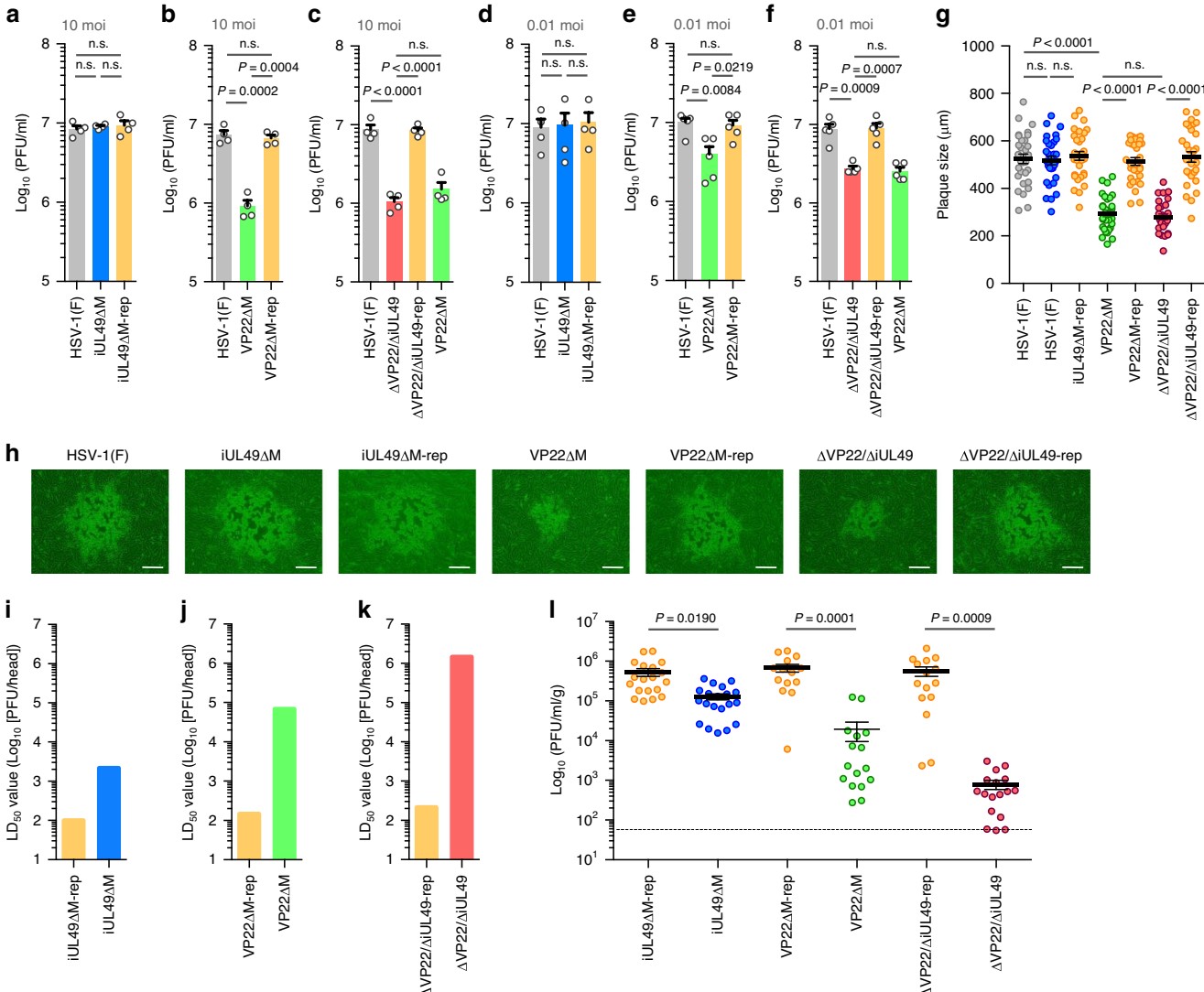

**Fig. 3 piUL49 is an HSV-1 virulence factor. a–f** Vero cells were infected with wild-type HSV-1(F) (**a–f**), iUL49ΔM (**a, d**), iUL49ΔM-rep (**a, d**), VP22ΔM (**b, c, e, f**), VP22ΔM-rep (**b, c, e, f**), ΔVP22/ΔiUL49 (**c, f**), or ΔVP22/ΔiUL49-rep (**c, f**) at an MOI of 10 (**a–c**) or 0.01 (**d–f**). Total virus titers in cell culture supernatants and infected cells at 21 (**a–c**) or 48 h (**d–f**) post-infection were assayed. Each value is the mean ± SEM of 4 (**a–c**) or 5 (**b–f**) independent experiments. The indicated P-values were obtained using one-way ANOVA followed by Tukey's test. n.s. not significant. **g, h** Thirty plaque diameters (**g**) and representative images (**h**) of Vero cells infected with each of the indicated viruses were measured at 48 h post-infection. Each value is the mean ± SEM for each group. The indicated P-values were obtained using one-way ANOVA followed by Tukey's test. n.s. not significant. Scale bar, 200 μm. **i–k** Three-week-old female ICR mice were infected intracranially with serial 10-fold dilutions of iUL49ΔM-rep (**i**), iUL49ΔM (**i**), VP22ΔM-rep (**j**), VP22ΔM (**j**), ΔVP22/ΔiUL49-rep (**k**), or ΔVP22/ΔiUL49 (**k**) in groups of six mice per dilution and monitored for 14 days. LD50 values were determined by the Behrens–Karber method. **l** Three-week-old female ICR mice were infected intracranially with 100 PFU/head of iUL49ΔM-rep, n = 20, iUL49ΔM, n = 19, VP22ΔM-rep n = 14, VP22ΔM, n = 16, ΔVP22/ΔiUL49-rep, n = 15 or ΔVP22/ΔiUL49, n = 17, and viral titers in the brains of infected mice at 3 day post-infection were assayed. Dashed line indicates the limit of detection. n.d. not detected. Each value is the mean ± SEM for each group. The indicated P-values were obtained using one-way ANOVA followed by Tukey's test. Source data are provided as a Source Data file.

mutation or the VP22ΔM mutation alone (Fig. 3i to l). These synergistic effects suggest that piUL49 and VP22 independently regulate HSV-1 virulence and replication in vivo.

**piUL49 binds to and activates HSV-1 dUTPase.** To determine the mechanism by which piUL49 regulates HSV-1 virulence and replication in vivo, we investigated the localization of MEF-piUL49 in HSV-1-infected cells. As shown in Fig. 4a, the pattern of localization of MEF-piUL49 in Vero cells infected with HSV-1 expressing MEF-piUL49 was similar to that of HSV-1 dUTPase (vdUTPase)[27]. Indeed, both co-localization (Fig. 4a) and physical proximity between MEF-piUL49 and vdUTPase (Supplementary

Fig. 11) in infected cells were detected. vdUTPase tagged with Strep-tag (SE-vdUTPase) specifically coprecipitated with piUL49 from lysates of Vero cells infected with HSV-1 expressing SE-vdUTPase (Fig. 4b and Supplementary Figs. 12 and 13). GST-vdUTPase201–280, but not GST alone or GST-vdUTPase281–371, efficiently pulled down Flag-piUL49 from lysates of HEK293FT cells ectopically expressing Flag-piUL49 (Supplementary Fig. 14). Collectively, these results indicate that piUL49 interacts with vdUTPase in HSV-1-infected cells. Furthermore, purified GST-vdUTPase201–280, but not purified GST alone or purified GST-vdUTPase281–371, efficiently pulled down purified SE-piUL49 (Supplementary Fig. 15). Reciprocally, purified GST-piUL49 efficiently pulled down purified 6xHis-SUMO-vdUTPase

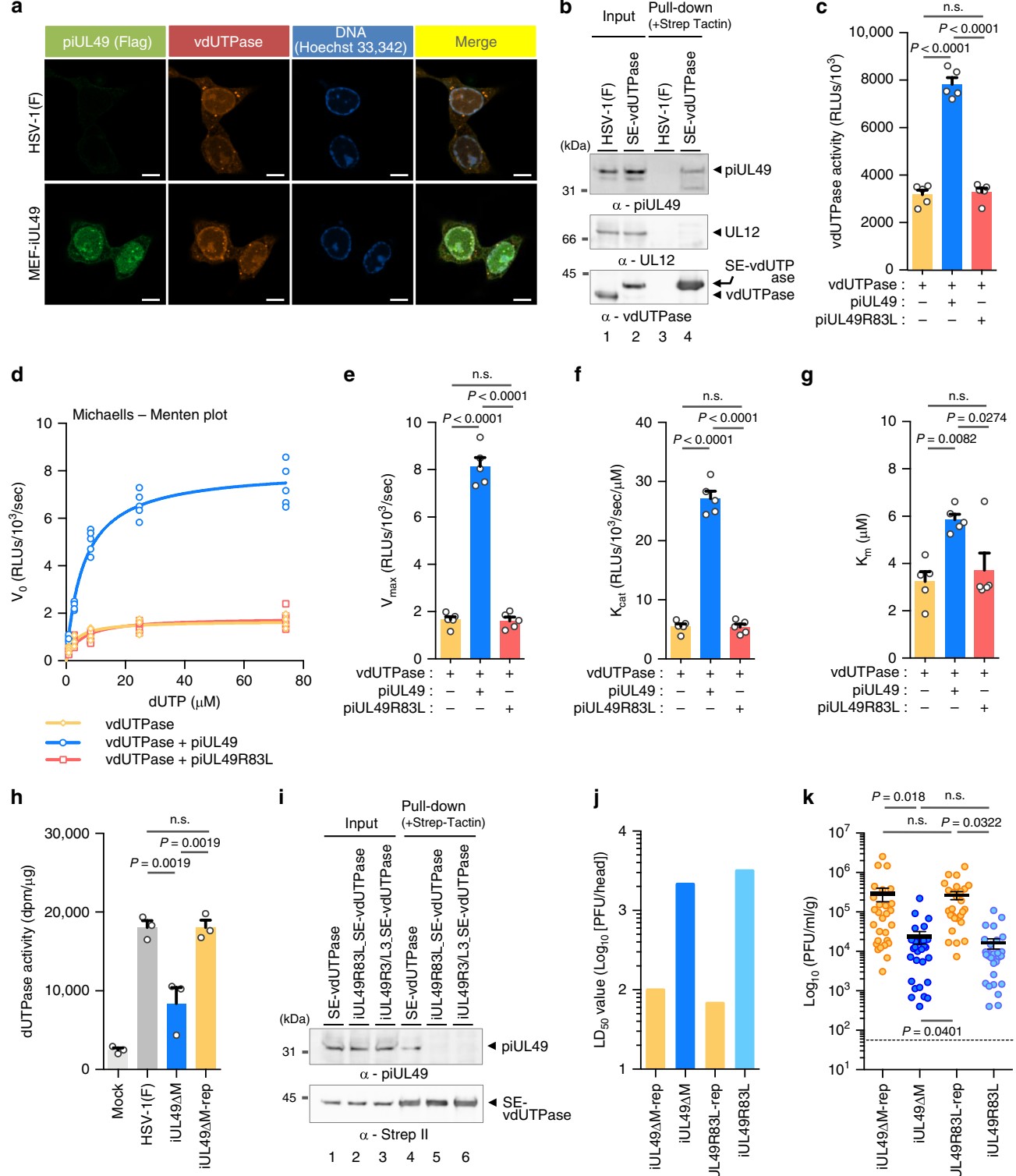

but not purified 6xHis-SUMO-vUNG (HSV-1 uracil-DNA gly-cosylase) (Supplementary Fig. 16). These results indicate that piUL49 directly binds to vdUTPase in vitro.

dUTPase is a key enzyme for the accurate replication of the DNA genome and is present in most organisms from prokaryotes to higher eukaryotes[28–31]. This enzyme catalyzes dUTP cleavage to dUMP and pyrophosphate (PP$_i$) to prevent dUTP misincor-poration into replicating DNA, which causes mutations and strand breakage, as well as providing a substrate for thymidylate synthase[28–31]. It was reported that vdUTPase is critical for HSV-1 replication and virulence when cellular endogenous dUTPase activity is low, such as in the brains of mice, by compensating for insufficient cellular dUTPase activity[32–34]. In an in vitro assay system to analyze vdUTPase activity using purified 6xHis-SUMO-vdUTPase and SE-piUL49 (Supplementary Fig. 17), piUL49 significantly increased the activity of vdUTPase under conditions where the molar ratio of the enzyme to piUL49 was 1:1 (Fig. 4c and Supplementary Figs. 18 and 19). Furthermore,

**Fig. 4 piUL49 interacts with vdUTPase and functions as an activator of the viral enzyme. a** Confocal microscope images of Vero cells infected with wild-type HSV-1(F) or MEF-iUL49 for 12 h at an MOI of 10 and stained with Hoechst33342 or antibodies to Flag and vdUTPase. Scale bar, 10 μm. **b** Vero cells were infected with HSV-1(F) or SE-vdUTPase for 12 h at an MOI of 10, lysed, precipitated with StrepTactin-sepharose and analyzed by immunoblotting with antibodies to iUL49, vdUTPase and UL12. **c** vdUTPase activity of 0.3 μM purified 6xHis-SUMO-vdUTPase or 6xHis-SUMO-vdUTPaseD97A with 74.07 μM dUTP in the absence or presence of 0.3 μM of purified SE-piUL49 or SE-piUL49R83L. The release of PPi upon dUTP hydrolysis by 6xHis-SUMO-vdUTPase was followed by measuring luciferase signaling and relative luminescence units (RLUs) were normalized by subtracting the value of vdUTPaseD97A from that of vdUTPase. Each value is the mean ± SEM of five independent experiments. The indicated $P$-values were obtained using one-way ANOVA followed by Tukey's test. n.s. not significant. **d**–**g** Kinetic analyses of vdUTPase activity using purified 6xHis-SUMO-vdUTPase and 6xHis-SUMO-vdUTPaseD97A in the absence or presence of purified SE-piUL49 or SE-piUL49R83L (**d**). $V_{max}$ values (**e**), $K_{cat}$ values (**f**) and $K_m$ values (**g**) based on (**d**) are shown. Each value is the mean ± SEM of five independent experiments. The indicated $P$-values were obtained using one-way ANOVA followed by Tukey's test. n.s. not significant. **h** dUTPase activities (dpm/μg of total protein) of Vero cells mock-infected and infected with HSV-1(F), iUL49ΔM or iUL49ΔM-rep for 12 h at an MOI of 10. Each value is the mean ± SEM of three independent experiments. The indicated $P$-values were obtained using one-way ANOVA followed by Tukey's test. n.s. not significant. **i** Vero cells infected with SE-vdUTPase, iUL49R83L_SE-vdUTPase, or iUL49R3/L3_SE-vdUTPase were lysed, precipitated with StrepTactin-sepharose, and analyzed by immunoblotting with antibodies to piUL49 and Strep II. **j**, **k** LD_50 values (**j**) or viral titers in the brains (**k**) ($n = 26$) of mice intracranially infected with iUL49ΔM-rep, iUL49ΔM, iUL49R3/L3, iUL49R83L-rep, or iUL49R83L were determined as described in Fig. 3i–l. The indicated $P$-values were obtained using one-way ANOVA followed by Tukey's test. n.s. not significant. Digital images are representative of three independent experiments (**a**, **b**, **i**). Source data are provided as a Source Data file.

compared with the effect of piUL49 on vdUTPase activity under conditions where the molar ratio of the enzyme to piUL49 was 1:1, insufficient addition of piUL49 in the assay system where the molar ratio of the enzyme to piUL49 was 1:0.33, significantly decreased the effect of piUL49 on vdUTPase activity. In contrast, excess addition of piUL49 where the molar ratio was 1:3 barely increased the effect of piUL49 on vdUTPase activity (Supplementary Fig. 18). Kinetic analyses using the assay system showed that the initial velocity ($V_0$) of PPi synthesis by vdUTPase fitted the Michaelis–Menten curve (Fig. 4d and Supplementary Fig. 19) and the maximum reaction velocity ($V_{max}$); furthermore, the catalytic turnover ($K_{cat}$) values of vdUTPase were enhanced by piUL49 (Fig. 4e, f and Supplementary Fig. 19). The $V_{max}$ and $K_m$ values were increased in parallel by piUL49 (Fig. 4e, g and Supplementary Fig. 19), indicating that piUL49 acted as an activator for vdUTPase by uncompetitive activation[35]. In agreement with these in vitro studies, the iUL49ΔM mutation significantly reduced dUTPase activity in HSV-1-infected Vero cells (Fig. 4h). Similar results were obtained with HSV-1-infected HFFF-2 and HaCaT cells (Supplementary Fig. 20a, b). Furthermore, the iUL49ΔM mutation had no effect on the accumulation and localization of vdUTPase, or the phosphorylation of vdUTPase at serine 187, which was shown to be critical for the enzymatic activity of vdUTPase[27] (Supplementary Figs. 20c–f, 21, and 22). These results suggest that piUL49 acts as an activator for vdUTPase and specifically regulates the enzymatic activity of vdUTPase by promoting the catalytic turn-over of vdUTPase in HSV-1-infected cells. Furthermore, an enzymatic-dead mutation in vdUTPase in combination with an iUL49ΔM mutation did not synergistically augment the LD_50 values in mice and reduce virus yields in brains compared with the enzyme-dead mutation in vdUTPase or the iUL49ΔM mutation alone (Supplementary Figs. 23 and 24). There results suggest that piUL49 regulates vdUTPase activity-dependent HSV-1 virulence and replication in vivo and supports our conclusion drawn from the cell culture experiments.

To investigate the relevance of the interaction between piUL49 and vdUTPase, we attempted to map the minimal amino acid(s) in iUL49 required for the interaction with vdUTPase. First, we generated a series of iUL49 mutant viruses wherein 6–12 positively charged amino acid in each of the 5 iUL49 domains were substituted with leucines (Supplementary Figs. 25 and 26a). Some blocks of mutations reduced the accumulation of piUL49 in infected cells (Supplementary Fig. 26a). However, leucine substitutions at iUL49 residues 75, 80, 83, 85, 87, 88, and 98 (iUL49R6H1/L7 mutations), and iUL49 residues 112, 117,

119–121, 129, and 136 (iUL49R7/L7 mutations) had no effect on piUL49 accumulation (Supplementary Fig. 26a). Furthermore, unlike WT iUL49, the iUL49R6H1/L7 mutant was not coprecipitated by SE-vdUTPase, whereas the iUL49R7/L7 mutant was coprecipitated by SE-vdUTPase as efficiently as WT iUL49 (Supplementary Fig. 26b, c). Therefore, we focused on iUL49 residues 75, 80, 83, 85, 87, 88, and 98, and generated additional recombinant viruses harboring leucine substitutions at iUL49 Arg-83, Arg-85, and Arg-87 (iUL49R3/L3) or Arg-83 alone (iUL49R83L; Supplementary Figs. 27 and 28a). Similar to the iUL49R6H1/L7 mutant, iUL49R3/L3 and iUL49R83L mutants were not coprecipitated by SE-vdUTPase (Fig. 4i). In contrast, the iUL49R83L, iUL49R3/L3, and iUL49R6H1/L7 mutants were coprecipitated by heat shock protein 70 (HSP70; Supplementary Fig. 28b), which was identified as a binding partner of piUL49 by the affinity purification of ectopically expressed Flag-piUL49 in HEK293 cells coupled with MS-based proteomics (Supplementary Table 4). The coprecipitation was as efficient as that of WT iUL49 (Supplementary Fig. 28b). These results indicate that piUL49 Arg-83 is specifically required for the interaction between piUL49 and vdUTPase in HSV-1-infected cells. Furthermore, purified GST-vdUTPase_201–280 did not pull down purified SE-piUL49R83L (Supplementary Fig. 15), and reciprocally, purified GST-piUL49R83L did not pull down purified 6xHis-SUMO-vdUTPase (Supplementary Fig. 16), indicating that piUL49 Arg-83 is also required for the direct binding of piUL49 to vdUTPase in vitro. In agreement with these results, the iUL49R83L mutation abolished the effect of piUL49 on vdUTPase in the in vitro assays (Fig. 4c–g and Supplementary Fig. 19), significantly reduced the dUTPase activity in HSV-1-infected Vero cells to levels similar to the iUL49ΔM mutation (Supplementary Figs. 29–31), and increased the LD_50 values (Fig. 4j) and reduced virus yields in brains to levels similar to the iUL49ΔM mutation (Fig. 4k). These results suggest that the interaction between piUL49 and vdUTPase is involved in the piUL49-mediated activation of vdUTPase and is critical for HSV-1 virulence and replication in vivo.

**piUL49 promotes to compensate for cellular dUTPase activity.** In sh-hDUT-HEp-2 cells, which were reported to show down-regulated expression and activity of cellular dUTPase[33] (Supplementary Fig. 32), the iUL49ΔM mutation significantly reduced progeny virus yields, but had no effect on control sh-Luc-HEp-2 cells (Fig. 5a, b). Furthermore, the ectopic expression of Flag-tagged cellular dUTPase (F-hDUTn) with a recombinant virus

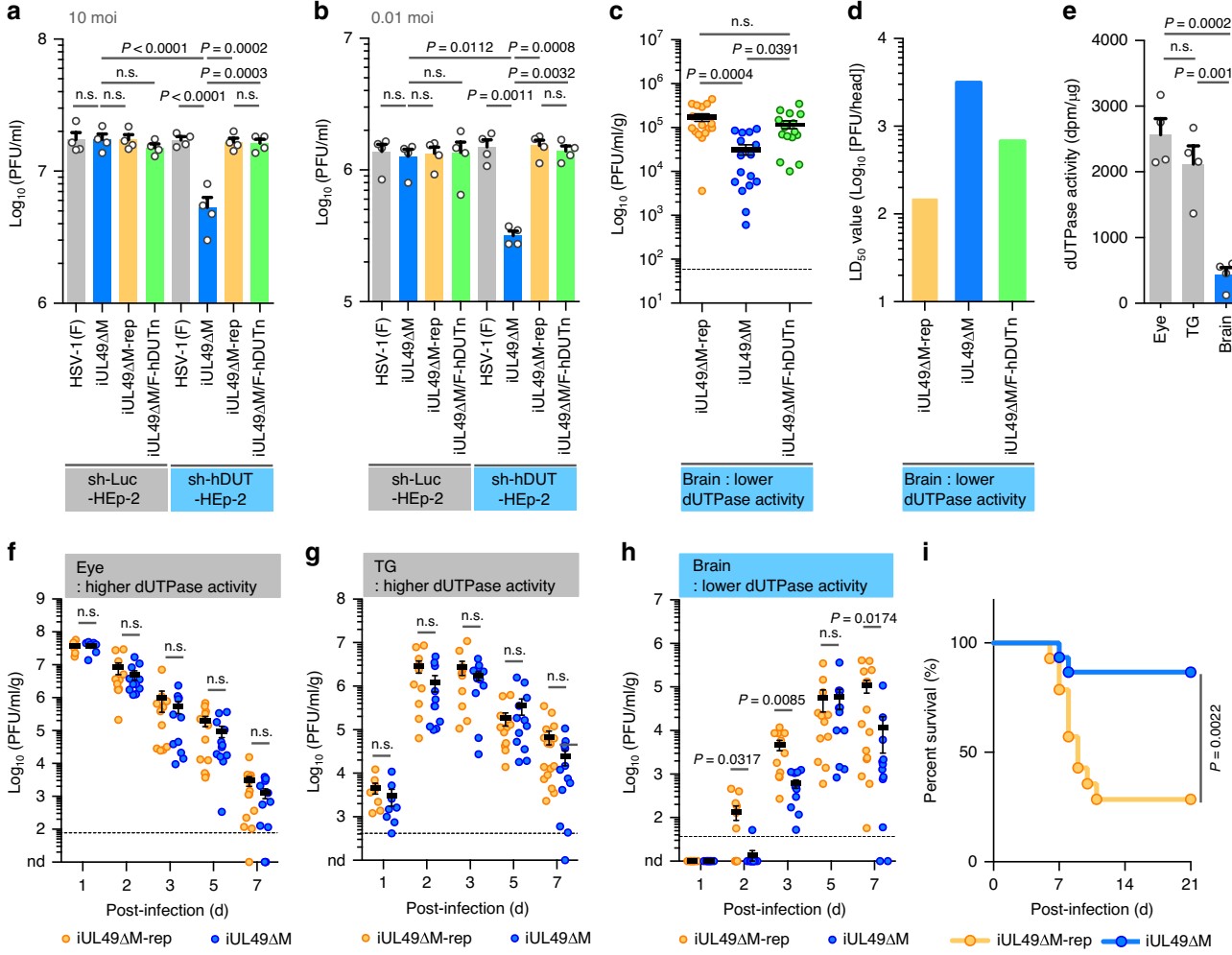

**Fig. 5 piUL49 is required to compensates for low cellular dUTPase activity. a, b** Viral titers of sh-Luc-HEp-2 or sh-hDUT-HEp-2 were infected with wild-type HSV-1(F), iUL49ΔM, iUL49ΔM-rep or iUL49ΔM/F-hDUTn for 36 h at an MOI of 10 (**a**) or for 60 h at an MOI of 0.01 (**b**), respectively, and assayed. Each value is the mean ± SEM of four independent experiments. The indicated $P$-values were obtained using one-way ANOVA followed by Tukey's test. n.s. not significant. **c, d** Viral titers in the brains (**c**) ($n = 16$) or LD$_{50}$ values (**d**) of 3-week-old female ICR mice intracranially infected with iUL49ΔM-rep, iUL49ΔM, or iUL49ΔM/F-hDUTn were determined as described in Fig. 3i–l. Each value is the mean ± SEM for each group. The indicated $P$-values were obtained using one-way ANOVA followed by Tukey's test. n.s. not significant. Dashed line indicates the limit of detection. **e** dUTPase activities (dpm/μg of total protein) in eyes, TGs and brains of nine 4-week-old female ICR mice. Each value is the mean ± SEM of four independent experiments. The indicated $P$-values were obtained using one-way ANOVA followed by Tukey's test. n.s. not significant. **f–h** Four-week-old female ICR mice were infected ocularly with $3 \times 10^6$ PFU/eye of iUL49ΔM or iUL49ΔM-rep. Viral titers in the eyes (**f**), TGs (**g**), or brains (**h**) of infected mice at 1 day ($n = 8$), 2 day ($n = 11$), 3 day ($n = 11$), 5 day ($n = 12$), and 7 day (iUL49ΔM-rep, $n = 16$; iUL49ΔM, $n = 12$) post-infection were assayed. Dashed line indicates the limit of detection. n.d. not detected. The indicated $P$-values were obtained using an unpaired two-tailed Student's $t$ test (eyes and TGs at 1 day, TGs at 2 day, and eyes and brains at 5 day) and Welch's $t$-test (eyes and brains at 2 day, eyes, TGs and brains at 3 and 7 day, and TGs at 5 day). n.s. not significant. **i** Four-week-old female ICR mice were infected ocularly with $3 \times 10^6$ PFU/eye of iUL49ΔM-rep ($n = 14$) or iUL49ΔM ($n = 15$) and monitored daily for survival for 21 days. The indicated $P$-values were obtained using Log-rank test. Source data are provided as a Source Data file.

iUL49ΔM/F-hDUTn expressing F-hDUTn (Supplementary Figs. 33 and 34) restored the impaired progeny virus yields with the iUL49ΔM mutation to the wild-type levels in sh-hDUT-HEp-2 cells (Fig. 5a, b). Similar results were also obtained with progeny virus yields in neuroblastoma SK-N-SH cells (Supplementary Fig. 35) and those in the brains of mice following intracranial HSV-1 infection (Fig. 5c). In agreement with these results, the iUL49ΔM mutation augmented the LD$_{50}$ values, and the ectopic expression of F-hDUTn in part restored the augmented LD$_{50}$ values with the mutation (Fig. 5d). In contrast, the iUL49ΔM mutation had no obvious effects on HSV-1 replication in the eyes or trigeminal ganglia (TGs), where endogenous cellular dUTPase activity and expression are relatively high compared with brains (Fig. 5e and Supplementary Fig. 36), of mice following ocular infection (Fig. 5f, g). Eventually, the iUL49ΔM mutation reduced HSV-1 replication in brains and mortality in mice following ocular infection similar to that observed in mice with intracranial infection (Fig. 5h, i). These results indicate that piUL49, like vdUTPase, is required for efficient viral replication and virulence when endogenous cellular dUTPase levels are low. Taken together, our series of data suggest that piUL49 interacts with vdUTPase and functions as an activator of this viral enzyme to compensate for insufficient cellular dUTPase activity and allow efficient viral replication and virulence.

## Discussion

Exploiting the property of HSV-1 infection whereby this virus shuts-off host cellular protein synthesis[3,10], we used chemical

proteomics integrating BONCAT and MS to identify previously cryptic orphan HSV-1 CDSs. We report nine previously unidentified CDSs whose translated products were expressed in HSV-1-infected cells. Although accumulating data based mainly on ribosome profiling in cells infected with various viruses have recently identified many putative previously unidentified viral ORFs[2,5–8], there is a lack of information regarding their biological significance in the context of viral infection. Therefore, it has been anticipated that the putative viral ORFs identified by this approach would not act as functional polypeptides during viral infection. In this study, we performed the in-depth characterization of one of these identified CDSs, designated piUL49, and showed that piUL49 was a critical HSV-1 neurovirulence factor that acts as an activator of a viral enzyme critical for accurate replication of the viral genome. Neurons and glia cells were reported to be the main targets of HSV-1 infection in the brains of mice,[36,37] suggesting piUL49 is likely to mediate its functions in these cell types. In agreement with this, we observed an effect of piUL49 in human neuroblastoma SK-N-SH cells but not in simian kidney epithelial Vero cells or human epithelioid carcinoma HEp-2 cells. Furthermore, the considerable conservation (in 80 HSV-1 strains) of most of the CDSs identified in this study suggests that viral polypeptides translated from these CDSs are likely to function in HSV-1 infection. Our study indicates the usefulness of chemical proteomics for the discovery of previously unidentified viral protein coding genes as well as demonstrating the necessity for the in-depth characterization of identified ORFs based on a comprehensive approach. A significant advantage of chemical proteomics, compared with ribosome profiling that detects translational status by monitoring ribosome occupancy on mRNAs[4], is to directly detect translated proteomes in infected cells. Because many viruses have evolved various mechanisms to shut-off host cellular protein synthesis[3,10,38–40], this chemical proteomic approach is widely applicable for the identification of unknown protein coding genes in many other viruses, especially, those of large and complex viruses.

In the genetic code, most amino acids are encoded by more than one codon, and for half of the 64 codons, the corresponding amino acid is independent of the third base in the codon. This degeneracy of the genetic code should reduce the deleterious effects of base substitutions at the third codon base allowing DNA to be more tolerant to point mutations. However, this genetic benefit is not effective when multiple genes are overlapped out of frame. Aa described above, viruses tend to densely pack coding information on their limited viral genomes. For example, in the HSV-1 genome, 42 of the 84 known HSV-1 genes and all the CDSs identified in this study, overlapped with other gene(s) out of frame(s). Indeed, a point mutation at the third codon base of the VP22 start codon completely precluded the expression of piUL49 in HSV-1 infected cells (Fig. 2b–d and Supplementary Figs. 3 and 4). Therefore, viruses including HSV-1 might have evolved mechanism(s) to replicate their genomes more accurately. Supporting this hypothesis, various DNA viruses including herpesviruses, poxviruses, adenoviruses, and African swine fever virus, as well as retroviruses encode dUTPase, which compensates for low cellular dUTPase activity if that situation is present in their host cells, e.g., in resting and differentiated cells such as neurons and macrophages[30,31,41]. In addition, some of these viruses including herpesviruses also encode uracil-DNA glycosylase (UNG) which, like dUTPase, has a critical role in the maintenance of genome integrity by preventing point mutations during genome replication[30,31,42]. Notably, HSV-1 appears to have evolved multiple mechanisms to regulate vdUTPase. These include the activator function of piUL49 for vdUTPase and the regulatory effect of phosphorylation mediated by HSV-1-encoded protein kinase Us3 on vdUTPase, both of which are critical for

viral replication and pathogenesis in vivo, as shown here and in earlier studies[32,33]. Such complex regulation has not been previously reported in cellular or other microbial dUTPases. These observations indicate unique mechanisms are involved in vdUTPase regulation and suggest that vdUTPase is necessary for tight regulation in infected cells and highlights the importance of vdUTPase in HSV-1 infection. Collectively, our study provides insights into the coevolution between the complication of viral genomes and the acquisition of viral functions for accurate replication of viral genomes.

## Methods

**Cells.** Simian kidney epithelial Vero cells[43] were provided by Dr Bernard Roizman and maintained in Dulbecco's modified Eagle's medium (DMEM) medium containing 5% calf serum. Human cervical epithelioid carcinoma HeLa (provided by Dr Shinobu Kitazume)[21], musculus macrophage J774.A1 (provided by Dr Yasunobu Yoshikai)[26], human keratinocyte HaCaT (purchased from the Cell Lines Service (CLS))[44], human Caucasian fetal foreskin fibroblast HFFF-2 (purchased from the European Collection of Authenticated Cell Cultures (ECACC)), human embryonic kidney epithelial HEK293FT (purchased from the Thermo Fisher Scientific), and HEK293 (purchased from the American Type Culture Collection (ATCC)) cells were maintained in DMEM containing 10% fetal calf serum (FCS). Human epithelioid carcinoma HEp-2 and human neuroblastoma SK-N-SH and rabbit skin cells (provided by Dr Bernard Roizman)[21] were maintained in DMEM containing 10% FCS. sh-Luc-HEp-2 and sh-hDUT-HEp-2 cells expressing short hairpin RNA (shRNA) against firefly luciferase or human dUTPase (hDUT)[33] were maintained in DMEM containing 10% FCS in combination with 50 μg/ml hygromycin B.

**Viruses.** Wild-type strains of HSV-1 [strain F: HSV-1(F), strain 17 syn+: HSV-1 (17), stain KOS: HSV-1(KOS), and strain VR3: HSV-1(VR3)] were provided by Dr Bernard Roizman, Dr Andrew J. Davison, Dr Yohei Yamauchi and Dr Tatsuo Suzutani[45–47]. In recombinant virus YK451 (ΔVP22), the UL49 gene encoding VP22 was disrupted by the insertion of a foreign gene cassette carrying a stop codon (TGA) just downstream of the UL49 start codon, and containing an I-SceI site, and a kanamycin resistance gene[24,26]. In YK750 (ΔvdUTPase), the UL50 gene encoding vdUTPase was disrupted by the insertion of a foreign gene cassette that contained an I-SceI site, a kanamycin resistance gene, and 37 bp of the vdUTPase sequence encoding codons 181 to 193 in which Ser-187 was replaced with alanine[27].

**Chemical proteomics.** HeLa cells were mock-infected or infected with HSV-1(F) at an MOI of 10. At the timepoints indicated, medium was removed and replaced with L-methionine-free Dulbecco's modified Eagle's medium (DMEM) (Sigma) containing 10% fetal calf serum (FCS) for 30 min to deplete methionine. Then, cells were labeled with L-methionine-free DMEM containing 50 μM AHA (L-azidohomoalanine; Molecular Probes) and 10% FCS from 0.5 to 4 h or 8 to 12 h post-infection, washed with ice-cold PBS twice, and lysed in 8 M urea buffer (8 M urea, 200 mM Tris [pH 8.4], 4% CHAPS, and 1 M NaCl) containing benzonase (Millipore) and protease inhibitor cocktail (Sigma). For click reactions, 100 μl of alkyne-conjugating agarose resin (Thermo Fisher Scientific), THPTA, CuSO₄, and L-ascorbic acid sodium salt were added according to the instructions of the Click-iT™ Protein Enrichment Kit (Thermo Fisher Scientific) and incubated for 12 h at room temperature while rotating. Resin was washed once with SDS wash buffer (100 mM Tris [pH8.4], 1% SDS, 250 mM NaCl, 5 mM EDTA), reduced with dithiothreitol, alkylated with iodoacetamide, then washed four times with SDS wash buffer, four times with 8 M urea in 100 mM Tris [pH 8.0], and four times with 20% acetonitrile. Washed resin was resuspended in 100 mM Tris [pH 8.0], 2 mM CaCl₂, and 10% acetonitrile, and digested with trypsin for 16 h at 37 °C with mixing. Digested peptides were recovered from the resin and cleaned up with a C18 mono-spin column (GL Science). Purified peptides were dissolved in 1% formic acid. Samples were analyzed using a nanoscale liquid chromatography-tandem mass spectrometry (LC-MS/MS) system as reported elsewhere[48]. The peptide mixture was applied to a Mightysil-PR-18 (Kanto Chemical) frit-less column (45 3 0.150 mm ID) and separated using a 0–40% gradient of acetonitrile containing 0.1% formic acid for 80 min at a flow rate 100 nL/min. Eluted peptides were sprayed directly into a mass spectrometer (Triple TOF 5600+; AB Sciex). MS and MS/MS spectra were obtained using the data-dependent mode, in which a 0.25-s TOF MS scan from 400–1500 m/z was performed, followed by 0.01-s production scans from 100 to 1500 m/z on the 25 most intense 2–3 charged ions. Centroid spectrum data were processed with Protein Pilot Software version 4.5 (Sciex). The Paragon algorithm (up to 5 missed cleavages can be searched) of Protein-Pilot was used to search RefSeq (NCBI) human protein database containing any potential amino acid sequences translated from HSV-1 genome reading frames and the tolerances were specified as ±0.05 Da for peptides and ±0.05 Da for MS/MS fragments. Peptides with an expectation value <0.44% false discovery rate were chosen for further data processing. For ProteinPilot™ searches, the following settings were selected: trypsin

specificity, Cys-carbamidomethylation. Processing parameters were set to "Biological modification".

**Quantification of protein or peptide concentrations.** Protein or peptide concentrations were determined using a Pierce BCA assay kit (Thermo Fisher Scientific) or DC Protein Assay (Bio-Rad) according to the manufacturer's instructions.

**Search for conservation of the HSV-1 CDSs identified in HSV-1 strains.** Each of the predicted peptide sequences of the CDSs identified in this study was compared with the corresponding CDSs of 80 HSV-1 strains (Supplementary Table 3) using BLAST and R[49].

**Plasmids.** pMAL-iUL49-P3 and pMAL-vUNG-P1, used to generate a fusion protein of maltose binding protein (MBP) and a domain of iUL49 and UL2, were constructed by cloning the domains of piUL49 (encoded by iUL49 codons 141–226) and vUNG (encoded by UL2 codons 1–100) amplified by PCR from pYEbac102[43] into pMAL-c (New England BioLabs) in frame with MBP. pGEX-vdUTPase-P1 and pGEX-vdUTPase-P2, used to generate a fusion protein of glutathione S-transferase (GST) and a domain of vdUTPase, were constructed by cloning the domain of vdUTPase encoded by UL50 codons 201–280 and codons 281–371, respectively, amplified from pYEbac102 by PCR[43] and cloning it into pGEX-4T-1 (GE Healthcare) in frame with GST. pE-SUMO-vdUTPase and pE-SUMO-vUNG, used to generate a SUMO fusion protein containing an N-terminal 6xHis-tag and a domain of vdUTPase and vUNG, were constructed by amplifying the entire coding sequences of vdUTPase (encoded by UL50) and vUNG (encoded by UL2) amplified from pYEbac102 by PCR[43], and cloning it into pE-SUMO-Amp (LifeSensors) linearized by *Bsa*I (New England Biolabs Japan) using the In-Fusion HD Cloning Kit (Takara) according to the manufacturer's instructions. pE-SUMO-vdUTPaseD97A encoding the SUMO fusion protein containing an N-terminal 6xHis-tag and a domain of vdUTPase with an alanine substituted for Asp-97 was constructed by amplifying the entire coding sequences of vdUTPaseD97A from vdUTPaseD97A genome by PCR, and cloning it into pE-SUMO-Amp linearized by *Bsa*I using the In-Fusion HD Cloning Kit.

pFlag-iUL49 was constructed by cloning the entire coding sequences of piUL49 encoded by iUL49 amplified from pYEbac102[43] by PCR into pFlag-CMV2 (Sigma) in frame with Flag. pcDNA-SE was constructed by the annealing oligonucleotides shown in Supplementary Table 5 and cloning it into *Hind*III and *Eco*RI sites of pcDNA3.1(+) (Thermo Fisher Scientific). To generate a fusion protein of Strep-tag and the entire coding sequences of iUL49, pcDNA-SE-iUL49 was constructed by cloning the entire coding sequences of piUL49 encoded by iUL49 amplified from pYEbac102[43] by PCR into pcDNA-SE in frame with a Strep-tag. pEu-GST-iUL49, used to generate a fusion protein of GST and a domain of iUL49 in the wheat germ cell-free protein expression system described below, was constructed by cloning the entire coding sequences of piUL49 encoded by iUL49 amplified from pYEbac102[43] by PCR[43], into pEu-E01-GST-TEV-MCS-N01 (CellFree Sciences) in frame with GST. To generate fusion proteins consisting of GST or a Strep-tag fused to piUL49 with a leucine substituted for Arg-83, pEu-GST-iUL49R83L, or pcDNA-SE-iUL49R83L, respectively, were constructed by amplifying the entire coding sequences of piUL49R83L from iUL49R83L genome by PCR, and cloning it into pEu-E01-GST-TEV-MCS-N01 and pcDNA-SE in frame with GST and a Strep-tag, respectively.

pBS-Venus-KanS, used to generate recombinant viruses in the two-step Red-mediated mutagenesis procedure described below, was constructed by amplifying the domain of pEP-KanS[50] carrying the I-*Sce*I site and the kanamycin resistance gene by PCR from pEP-KanS using the primers shown in Supplementary Table 5 and cloning it into the *Pst*I site of pBS-Venus[21]. piUL49R9H3/L12-KanS-pst+, piUL49R6H1/L7-KanS-pst+, piUL49R7/L7-KanS-pst+, piUL49R5H1/L6-KanS-pst+, or piUL49R7H1/L8-KanS-pst+ used to generate recombinant viruses by the two-step Red-mediated mutagenesis procedure described below, were constructed by cloning the domain of pEP-KanS[50] carrying the I-*Sce*I site and the kanamycin resistance gene by PCR from pEP-KanS using the primers shown in Supplementary Table 5, into the *Pst*I sites of pUC57-iUL49R9H3/L12-pst+, pUC57-iUL49R6H1/L7-pst+, pUC57-iUL49R7/L7-pst+, pUC57-iUL49R5H1/L6-pst+, or pUC57-iUL49R7H1/L8-pst+, respectively, which were synthesized (GenScript). The synthesized DNA sequences are shown in Supplementary Table 6.

pRB-EGRp-Flag-hDUTn-polyA-KanS, used to generate a recombinant virus in the two-step Red-mediated mutagenesis procedure described below, was constructed by cloning the domain of human cellular dUTPase isoform 1 (hDUTm) encoded by hDUT-M codons 95–252, the I-*Sce*I site and the kanamycin resistance gene, amplified by PCR from pBS-hDUTm-KanS[33] using the primers shown in Supplementary Table 5, into the *Spe*I and *Eco*R1 site of pRB5160, which contained the Egr-1 promoter region, a multicloning site, and bidirectional polyadenylation signals of HSV-1 UL21 and UL22 genes[51].

**Construction of recombinant viruses.** Recombinant viruses Venus-iUL49 or MEF-iUL49 expressing iUL49 fused to a Venus or MEF (for Myc tag, the Tobacco Etch Virus protease cleavage site, and Flag) tag[52], Venus-iUL44b and SE-vdUTPase expressing UL50 (vdUTPase) fused to a Strep-tag (Supplementary Figs. 3 and 12), were generated by the two-step Red-mediated mutagenesis procedure[27,50] using

*Escherichia coli* (*E.coli*) GS1783 containing pYEbac102Cre, a full-length infectious HSV-1(F) clone[43,53], pGEM-MEF[52], pBS-Venus-KanS, pEP-KanS-SEM[54], and the primers listed in Supplementary Table 5. Recombinant viruses Venus-iUL49ΔM and MEF-iUL49ΔM, in which the start codons in Venus-iUL49 and MEF-iUL49 were replaced with threonine, respectively (Supplementary Figs. 3 and 12), were generated by the two-step Red-mediated mutagenesis procedure[27,50] using *E. coli* GS1783 containing the Venus-iUL49 or MEF-iUL49 genome and the primers listed in Supplementary Table 5. Recombinant viruses iUL49ΔM or VP22ΔM, in which the iUL49 or UL49 (VP22) genes were inactivated by replacing the start codon of iUL49 with threonine or the start codon of VP22 with a stop codon, respectively, and ΔVP22/ΔiUL49, in which the UL49 (VP22) and iUL49 genes were disrupted by deleting VP22 codons 1–266 with a kanamycin resistance gene (Supplementary Fig. 3), were generated by the two-step Red-mediated mutagenesis procedure[27,50] using *E. coli* GS1783 containing pYEbac102Cre[53], and the primers listed in Supplementary Table 5. Recombinant viruses iUL49ΔM-rep or VP22ΔM-rep, in which null-mutations in iUL49ΔM or VP22ΔM, respectively, were repaired (Supplementary Figs. 3 and 8), were generated by the two-step Red-mediated mutagenesis procedure[27,50] using *E. coli* GS1783 containing the iUL49ΔM or VP22ΔM genomes and the primers listed in Supplementary Table 5. Recombinant virus ΔVP22/ΔiUL49-rep, in which the null-mutations in ΔVP22/ΔiUL49 were repaired (Supplementary Fig. 8), was generated by the two-step Red-mediated mutagenesis procedure[27,50] using *E. coli* GS1783 containing the ΔVP22/ΔiUL49 genomes, the primers listed in Supplementary Table 5 and VP22ΔM genomes carrying the kanamycin resistance gene inserted into the UL49 locus.

The recombinant virus vdUTPaseD97A, in which the enzymatic activity of the vdUTPase gene was inactivated by replacing aspartic acid at residue Asp-97 with an alanine as reported previously[32,33,55], or iUL49ΔM/vdUTPaseD97A, in which the expression of iUL49 and the enzymatic activity of the vdUTPase gene were inactivated by replacing the start codon of iUL49 with threonine and Asp-97 of vdUTPase with an alanine (Supplementary Fig. 23), were generated by the two-step Red-mediated mutagenesis procedure[27,50] using *E. coli* GS1783 containing pYEbac102Cre[53] or iUL49ΔM. The primers used are listed in Supplementary Table 5. Recombinant viruses vdUTPaseDA-rep or iUL49ΔM/vdUTPaseDA-rep in which D97A mutations in vdUTPase were repaired (Supplementary Fig. 23), were generated by the two-step Red-mediated mutagenesis procedure[27,50] using *E. coli* GS1783 containing the vdUTPaseD97A or iUL49ΔM/vdUTPaseD97A genomes. The primers used are listed in Supplementary Table 5.

The recombinant viruses iUL49R9H3/L12_SE-vdUTPase, iUL49R6H1/L7_SE-vdUTPase, iUL49R7/L7_SE-vdUTPase, iUL49R5H1/L6_SE-vdUTPase, or iUL49R7H1/L8_SE-vdUTPase, in which the arginine and/or histidine codons in iUL49 were replaced with leucine codons, and UL50 (vdUTPase) fused to a Strep-tag (Supplementary Fig. 25), were generated as follows. VP22 codons 1–266 in the SE-vdUTPase genomes were replaced with a zeocin resistance gene (ΔiUL49/ΔVP22/zeo+/SE-vdUTPase) by the two-step Red-mediated mutagenesis procedure[27,50] using *E. coli* GS1783 containing SE-vdUTPase, pEM7/zeo (Thermo Fisher Scientific). The primers used are listed in Supplementary Table 5. Then, a zeocin resistance gene in the ΔiUL49/ΔVP22/zeo+/SE-vdUTPase genome was replaced with sequences containing the mutant iUL49 genes (iUL49R9H3/L12-pst + _SE-vdUTPase, iUL49R6H1/L7-pst + _SE-vdUTPase, iUL49R7/L7-pst + _SE-vdUTPase, iUL49R5H1/L6-pst + _SE-vdUTPase, or iUL49R7H1/L8-pst + _SE-vdUTPase) by the two-step Red-mediated mutagenesis procedure[27,50] using *E. coli* GS1783 containing the piUL49R9H3/L12-KanS-pst+, piUL49R6H1/L7-KanS-pst+, piUL49R7/L7-KanS-pst+, piUL49R5H1/L6-KanS-pst+, or piUL49R7H1/L8-KanS-pst+. The primers used are listed in Supplementary Table 5. Finally, the artificially introduced *Pst*I restriction site in the UL49 locus was removed (iUL49R9H3/L12_SE-vdUTPase, iUL49R6H1/L7_SE-vdUTPase, iUL49R7/L7_SE-vdUTPase, iUL49R5H1/L6_SE-vdUTPase, or iUL49R7H1/L8_SE-vdUTPase) by the two-step Red-mediated mutagenesis procedure[27,50] using *E. coli* GS1783 containing the iUL49R9H3/L12-pst + _SE-vdUTPase, iUL49R6H1/L7-pst + _SE-vdUTPase, iUL49R7/L7-pst + _SE-vdUTPase, iUL49R5H1/L6-pst + _SE-vdUTPase, or iUL49R7H1/L8-pst + _SE-vdUTPase genomes. The primers used are listed in Supplementary Table 5.

The recombinant virus iUL49R83L_SE-vdUTPase, in which Arg-83 in iUL49 was substituted with a leucine and UL50 (vdUTPase) was fused to a Strep-tag, or the recombinant virus iUL49R3/L3_SE-vdUTPase, in which Arg-80, Arg-83, and Arg-85 in iUL49 were substituted with leucines and UL50 (vdUTPase) was fused to a Strep-tag (Supplementary Fig. 27), were generated by the two-step RRed-mediated mutagenesis procedure[27,50] using *E. coli* GS1783 containing SE-vdUTPase and the primers listed in Supplementary Table 5.

The recombinant virus iUL49R83L, encoding iUL49 with a leucine substituted for Arg-83, or iUL49R3/L3, encoding iUL49 with a leucine substituted for each of Arg-80, Arg-83, and Arg-85 (Supplementary Fig. 29), were generated by the two-step Red-mediated mutagenesis procedure[27,50] using *E. coli* GS1783 containing pYEbac102Cre[53] and the primers listed in Supplementary Table 5. Recombinant viruses iUL49R83L-rep or iUL49R3/L3-rep harboring a leucine mutation(s) in iUL49 were repaired by the two-step Red-mediated mutagenesis procedure[27,50] using *E. coli* GS1783 containing the iUL49R83L or iUL49R3/L3 genomes. The primers used are listed in Supplementary Table 5.

Recombinant virus ΔiUL49/F-hDUTn, in which the iUL49 gene was inactivated by threonine replacement in the start codon of iUL49 and an expression cassette consisting of the Egr-1 promoter, the human cellular dUTPase isoform 2 (hDUTn)

fused to Flag (F-hDUTn), and bidirectional polyadenylation signals of the HSV-1 UL21 and UL22 genes into the intergenic region between the UL50 and UL51 genes (Supplementary Fig. 33), was generated by the two-step Red-mediated mutagenesis procedure[27,50] using *E. coli* GS1783 containing pYEbac102Cre[53], pRB-EGRp-Flag-hDUTn-polyA-KanS, and the primers listed in Supplementary Table 5. The genotype of each recombinant virus was confirmed by PCR or sequencing.

**Production and purification of GST and MBP fusion proteins**. GST-fusion proteins GST-vdUTPase$_{201-280}$ and GST-vdUTPase$_{281-371}$, and MBP fusion proteins MBP-piUL49-P3 and MBP-vUNG-P1, were expressed in *E. coli* Rosetta (Novagen) or BL21 Star™ (DE3; Thermo Fisher Scientific) that were transformed with pGEX-vdUTPase-P1, pGEX-vdUTPase-P2 pMAL-iUL49-P3, or pMAL-vUNG-P1. Plasmid-containing bacterial cultures were grown at 37 °C with shaking in Luria-Bertani (LB) medium containing 100 μg/mL of ampicillin. Isopropyl-β-D-thiogalactopyranoside (IPTG) was added into the culture to a final concentration of 0.1 mM for GST-fusion protein or 0.3 mM for MBP-fusion protein when the culture reached an absorbance of ~0.4 at 600 nm. The culture was incubated at 37 °C for at least 2 h and then harvested by centrifugation at $9650 \times g$ for 5 min at 4 °C. After PBS washed with PBS, and stored at −20 °C. Cell lysates were prepared by re-suspending the pellets in GST buffer (25 mM Tris-HCl [pH 8.0], 500 mM NaCl, 10 mM 2-Mercaptoethanol) containing 1% Triton X-100 and PBS containing 1% Tween 20 by sonicating briefly on ice. They were then clarified by centrifugation twice at $25,000 \times g$ for 10 min at 4 °C and the supernatants were pre-cleared by incubation with glutathione-sepharose beads (GE Healthcare Life Science) or amylose beads (New England Biolabs Japan) at 4 °C for 2 h. After a brief centrifugation, sepharose beads were collected by brief centrifugation, washed extensively with GST buffer or PBS. The MBP fusion proteins were eluted with MBP elution-buffer (50 mM Tris-HCl [pH 8.0], 25 mM EGTA, 10 mM D (+)-Maltose Monohydrate) and stored at −80 °C.

**Antibodies**. Antibodies were purchased and used as follows: commercial mouse monoclonal antibodies to ICP27 (8. F. 1378; Virusys; 1:1000), glycoprotein B (gB) (H1817; Virusys; 1:1000), UL42 (13C9; Santa Cruz Biotechnology; 1:1000), human dUTPase (F6; Santa Cruz Biotechnology; 1:50), ICP8 (10A3; Millipore; 1:1000), Strep-tag (4F1; MBL; 1:1000), His-tag (OGHis; MBL; 1:1000), Flag-tag (M2; Sigma; 1:1000), biotin-anti-human-IL1β (13-7112-81; eBioscience; 1:500) and β actin (AC15; Sigma; 1:1000); rabbit polyclonal antibodies against green fluorescent protein (GFP) (598; MBL; 1:1000), early endosome antigen 1 (EEA1) (ab2900; Abcam; 1:100) and heat shock protein 70 (ab79852; Abcam; HSP70; 1:100); armenian hamster monoclonal antibody to anti-mouse/rat-IL-1β (15257977; eBioscience; 1:500), and goat polyclonal antibody to HSV-1 thymidine kinase (vTK) (vL-20; Santa Cruz Biotechnology; 1:500). To generate mouse polyclonal antibodies to HSV-1 piUL49 and vUNG, BALB/c mice were immunized once with purified MBP-piUL49-P3 or MBP-vUNG-P1, respectively, and TiterMax Gold (TiterMax USA, Inc.). Sera from immunized mice were used as sources of mouse polyclonal antibodies to piUL49 (1:100) and vUNG (1:100). To generate rabbit polyclonal antibodies to HSV-1 piUL49 (1:1000), rabbits were immunized with purified MBP-piUL49-P3, following the standard protocol of Eurofins Genomics (Tokyo, Japan). Rabbit polyclonal antibodies to VP16 (1:1000), VP22 (1:2000), VP23 (1:2000), or UL12 (1:2000) were reported previously[24,56,57].

**Immunoblotting**. Cell lysates in SDS sample buffer (62.5 mM Tris-HCl [pH 6.5], 20% glycerol, 2% SDS, 5% 2-mercaptoethanol) were sonicated, heated at 100 °C and subjected to electrophoresis in denaturing gels, transferred to nitrocellulose membranes (Bio-Rad). The membranes were blocked with 5% skim milk in T-PBS (phosphate-buffered saline (PBS) containing 0.05% Tween 20) for 30 min and reacted with indicated antibodies at least 2 h at room temperature or 4 °C. These membranes were then reacted with secondary antibodies conjugated with perox-idase (GE Healthcare Bio-Sciences) at least 1 h at room temperature or 4 °C, and visualized using ECL (GE Healthcare Bio-Sciences) with ImageQuant LAS 4000 (GE Healthcare Bio-Sciences). The amount of protein present in immunoblot bands was quantified using the ImageQuant LAS 4000 with ImageQuant TL7.0 analysis software (GE Healthcare Life Sciences) according to the manufacturer's instructions.

**Immunofluorescence**. Vero cells cultured on 35-mm-diameter glass-bottom dishes (Matsunami) were infected with wild-type HSV-1(F) or each of the recombinant viruses e at an MOI of 10. At 12 h post-infection, infected cells were fixed with 2% paraformaldehyde for 10 min, permeabilized with 0.1% Triton-X100 for 5 min, and blocked with PBS containing 10% human serum (Sigma-Aldrich) in PBS for at least 30 min at room temperature or overnight at 4 °C. These cells were reacted with indicated antibodies for 2 h at room temperature, followed by reaction with secondary antibodies conjugated to Alexa Fluor (Invitrogen) for 1 h at room temperature. Then, the cells were examined with a Zeiss LSM800 microscope (Zeiss, Oberkochen, Germany). Hoechst 33342 (Thermo Fisher Scientific; 1:2000) was used for the specific staining of the nuclei of fixed cells.

**Determination of plaque size**. Vero cells were infected with wild-type HSV-1(F) or each of the recombinant viruses under plaque assay conditions. After adsorption

for 1 h, the inoculum was removed, and the cell monolayers were overlaid with medium 199 (Sigma) containing 1% FCS and 160 μg of pooled human immu-noglobulin (Sigma)/ml. At 48 h post-infection, 30 plaques produced by HSV-1(F) or each of the recombinant viruses were analyzed using a Olympus IX73 micro-scope equipped with a digital DP80 camera (Olympus) and cellSens software (Olympus).

**Proximity ligation assay**. A Duolink kit was purchased from Sigma and assays were performed according to the manufacturer's instructions (Olink Bioscience). Vero cells grown on 35-mm diameter glass-bottomed dishes (Matsunami) were infected with wild-type HSV-1(F) or MEF-iUL49 at an MOI of 10. At 12 h post-infection, the cells were fixed with 2% paraformaldehyde for 10 min, permeabilized with 0.1% Triton X-100 for 2 min, and blocked with PBS containing 10% human serum for 30 min followed by incubation with primary antibodies to vdUTPase (rabbit polyclonal) and flag (mouse monoclonal) for 2 h at room temperature. After primary staining, these cells were washed twice with PBS and once with Duolink buffer A (10 mM Tris [pH 7.4], 150 mM NaCl, and 0.05% Tween 20), and reacted with secondary anti-rabbit antibodies conjugated with minus and anti-mouse conjugated with plus Duolink in situ proximity ligation assay (PLA) probes for 1 h at 37 °C. Then, the cells were washed twice with Duolink buffer A and incubated with ligation-solution for 30 min at 37 °C followed by washing twice with Duolink buffer A and subsequent incubation with amplification-polymerase solution for 100 min at 37 °C containing Duolink In Situ Detection Reagents Red. Finally, the cells were washed twice with Duolink buffer B (200 mM Tris [pH 7.5], 100 mM NaCl) and once with Duolink buffer C (2 mM Tris [pH 7.5], 1 mM NaCl), and examined with a Zeiss LSM800 microscope (Zeiss).

**Animal studies**. Female ICR mice were purchased from Charles River. Each of the recombinant viruses were diluted in medium 199 (Sigma) containing 1% FCS. For intracranial infection, 3-week-old mice were infected intracranially with 50 μl of 10-fold serial dilutions of the indicated viruses (6 mice per dilution) using a 27-gauge needle (TOP) to penetrate the scalp and cranium over the hippocampal region of the left hemisphere with a needle guard to prevent penetration further than 3 mm[43,53,58]. Mice were monitored daily, and mortality from 1 to 14 day post-infection was attributed to the infected virus. LD$_{50}$ values were calculated by the Behrens–Karber method.

For ocular infection, corneas of 4-week-old mice were lightly scarified with a 27-gauge needle (Terumo), the tear film was blotted, and a 4-μl drop containing $3 \times 10^6$ PFU/eye of the indicated virus was applied to the eye. Then, infected mice were monitored daily until 21 day post-infection. Infected mice were euthanized at the indicated timepoints after infection, and eyes, trigeminal ganglia, and/or brains were removed, sonicated in 0.5 mL Medium 199 containing 1% FCS and antibiotics, and frozen at −80 °C. Frozen samples were later thawed and centrifuged, after which viral titers in the supernatants were determined by standard plaque assays on Vero cells. All animal experiments were carried out in accordance with the Guidelines for Proper Conduct of Animal Experiments, Science Council of Japan. The protocol was approved by the Institutional Animal Care and Use Committee (IACUC) of the Institute of Medical Science, The University of Tokyo (IACUC protocol approval number: 16–69 and 11–81).

**Strep-tag affinity precipitation**. Vero cells were infected with wild-type HSV-1 (F), recombinant virus SE-vdUTPase, iUL49R6H1/L7_SE-vdUTPase, iUL49R7/L7_SE-vdUTPase, iUL49R83L_SE-vdUTPase, and/or iUL49R3/L3_SE-vdUTPase at an MOI of 10. Infected cells were harvested at 12 h post-infection and lysed in 0.5% NP-40 buffer (50 mM Tris-HCl [pH 8.0], 150 mM NaCl, 50 mM NaF, and 0.5% Nonidet P-40 [NP-40]) containing a protease inhibitor cocktail (Nacalai Tesque). Supernatants obtained after centrifugation of the cell lysates were pre-cleared by incubation with protein A-Sepharose beads (GE Healthcare) at 4 °C for 30 min. After a brief centrifugation, supernatants were reacted at 4 °C for 1.5 h with Strep-Tactin sepharose beads (IBA Lifescience). Sepharose beads were collected by brief centrifugation, washed extensively with 0.5% NP-40 buffer, and analyzed by immunoblotting with antibodies to piUL49, UL12, vdUTPase, and/or Strep-tag.

**Production and purification of His fusion proteins**. The expression plasmid pE-SUMO-vdUTPase, pE-SUMO-vdUTPaseD97A, and pE-SUMO-vUNG were used to transform *E. coli* Rosetta (Novagen). Plasmid-containing bacterial cultures were grown at 28 °C with shaking in Luria-Bertani (LB) medium containing 100 μg/mL of ampicillin. Isopropyl-β-D-thiogalactopyranoside (IPTG) was added into the culture to a final concentration of 0.1 mM when the culture reached an absorbance of ~0.3 at 600 nm. The culture was incubated at 16 °C overnight and then harvested by centrifugation at $9250 \times g$ for 5 min at 4 °C, washed with PBS, and stored at −80 °C. Cell lysates were prepared by re-suspending the pellets in xTractor Buffer (Takara) and by sonicating briefly on ice. They were then clarified by centrifugation at $88,600 \times g$ for 20 min at 4 °C and the supernatant filtered through a 0.45-μm filter (Sartorius) prior to loading onto a Capturem™ His-Tagged Purification Miniprep Kit (Takara). After unbound proteins were washed away with two washes of wash buffer (20 mM Na$_3$PO$_4$, 150 mM NaCl, and pH 7.6) and once with wash buffer containing 40 mM imidazole, the target protein was eluted with elution-buffer A (20 mM Na$_3$PO$_4$, 500 mM NaCl, 500 mM imidazole, and pH 7.6). The eluted

fraction was dialyzed twice with a desalting column (Apro Science) according to the manufacturer's instructions and stored at −80 °C.

**In vitro transcription and translation**. A WEPRO7240G expression kit was purchased from CellFree Sciences, and in vitro transcription and translation were performed according to the manufacturer's instructions. pEu-GST-iUL49 and pEu-GST-iUL49R83L were used for in vitro transcription with SP6 RNA polymerase, and the protein translation was performed at 16 °C for 20 h in a bilayer compound. One of the layers had a liquid mixture of mRNA, creatine kinase, and wheat germ extract WEPRO7240G; the other layer had translation buffer SUB-AMIX SGC. The translation reaction mixture was diluted with PBS, incubated with glutathione-sepharose beads (GE Healthcare Life Science) at 4 °C for 2 h, and the resin was washed four times with PBS. The target protein was eluted with elution-buffer B (50 mM Tris, 10 mM reduced-glutathione, pH 8.0). The eluted was dialyzed twice with a desalting column (AproScience) according to the manufacturer's instructions and stored at −80 °C.

**Production and purification of strep-tag fusion proteins**. HEK293FT cells were transfected with pcDNA-SE-iUL49 or pcDNA-SE-iUL49R83L using PEI Max. At 12 h post-transfection, transfected HEK293FT cells were infected with a recombinant virus YK750 (ΔvdUTPase)[27] at an MOI of 1 and harvested at 24 h post-infection. The cells were then lysed in radioimmunoprecipitation assay buffer (50 mM Tris-HCl [pH 7.5], 150 mM NaCl, 1% Nonidet P-40 [NP-40], 0.5% deoxycholate, 0.1% sodium dodecyl sulfate) containing a protease inhibitor cocktail (Nacalai Tesque). Supernatants obtained after centrifugation of the lysates at 88,600 × $g$ for 20 min at 4 °C were precleared by incubation with protein G-sepharose beads (GE Healthcare) at 4 °C for 30 min. After a brief centrifugation, supernatants were reacted with Strep-Tactin sepharose beads (IBA Lifescience) for 4 h at 4 °C. Sepharose beads were collected by a brief centrifugation and washed once with high-salt buffer (1 M NaCl, 10 mM Tris-HCl [pH 8.0], 0.2% NP-40), twice with low-salt buffer (0.1 M NaCl, 10 mM Tris-HCl [pH 8.0], 0.2% NP-40), eight times with radioimmunoprecipitation assay buffer, and finally four times with PBS. The target protein was eluted with elution-buffer C (100 mM Tris-HCl [pH 8.0], 150 mM NaCl, 1 mM EDTA, 10 mM D-desthiobiotin). The eluted was dialyzed twice with a desalting column (AproScience) according to the manufacturer's instructions and stored at −80 °C.

**GST pull-down assay**. HEK293FT cells were transfected with pFLAG-iUL49 using PEI Max and harvested at 48 h post-transfection. Cells were then lysed in 0.5% NP-40 buffer (50 mM Tris-HCl [pH 8.0], 150 mM NaCl, 50 mM NaF, and 0.5% Nonidet P-40 [NP-40]) containing a protease inhibitor cocktail (Nacalai Tesque). GST, GST-vdUTPase$_{201-280}$, and GST-vdUTPase$_{281-371}$ were expressed in E. coli Rosetta (Novagen) that had been transformed with pGEX4T-1, pGEX4T-vdUT-Pase-P1, and pGEX4T-vdUTPase-P2, respectively, and purified using glutathione-sepharose beads (GE Healthcare). The transfected cell lysates were reacted with purified GST proteins immobilized on glutathione-sepharose resin for 1 h at 4 °C. After extensive washing of the resin with 0.5% NP-40 buffer, the resin was divided into two parts. One part was analyzed by immunoblotting with anti-Flag antibody and the other was electrophoretically separated in a denaturing gel and stained with Coomassie brilliant blue (CBB). In other studies, GST, GST-vdUTPase$_{201-280}$ and GST-vdUTPase$_{281-371}$ expressed in E. coli Rosetta (Novagen) or GST-piUL49 and GST-piUL49R83L translated in vitro using a WEPRO7240G expression kit (Cell-Free Sciences) were purified using glutathione-sepharose beads (GE Healthcare). Then, purified 6xHis-SUMO-vdUTPase, 6xHis-SUMO-vUNG, SE-piUL49, or SE-piUL49R83L were reacted with purified GST proteins immobilized on glutathione-sepharose resin for 1 h at 4 °C. After extensive washing of the resin with 0.5% NP-40 buffer, the resin was analyzed by immunoblotting with anti-Flag, anti-piUL49, or anti-His-tag antibodies, electrophoretically separated in a denaturing gel, and stained with CBB.

**dUTPase enzyme assay**. Vero, HFFF-2, or HaCaT cells were mock-infected or infected with wild-type HSV-1(F) and/or each recombinant virus at an MOI of 10. Infected cells were harvested at 12 h post-infection, and solubilized in NP-40 buffer (50 mM Tris-HCl [pH 8.0], 150 mM NaCl, and 0.5% NP-40). Protein concentrations in the supernatants obtained after a brief centrifugation were determined using a Bio-Rad protein assay kit. Then, 10 μg of each supernatant was mixed with 200 μl reaction buffer (50 mM Tris-HCl [pH 8.0], 2 mM β-mercaptoethanol, 1 mM MgCl$_2$, 0.1% bovine serum albumin, 2 mM p-nitrophenylphosphate, and 74.07 μM dUTP [GeneAct] and 1.0 μCi [$^3$H]dUTP [Moravek, Inc.]). The reaction was allowed to proceed for 30 min at 37 °C and then terminated by spotting the reaction mixture onto DE81 circle disks (Whatman). The disks were washed three times for 5 min each with washing solution (1 mM ammonium formate and 4 M formic acid), followed by one wash with 95% ethanol for 3 min. The disks were air-dried and counted for radioactivity using a Liquid scintillation counter LSC-5100 (Aloka). In other studies, eyes, TGs, or brains were removed from 4-week-old female ICR mice (Charles River), washed with PBS, and sonicated in NP-40 buffer. After a brief centrifugation, dUTPase enzymatic activities (dpm/μg of total protein) were measured as described above.

**Kinetic analysis of vdUTPase enzyme activity**. The vdUTPase-produced PPi was monitored with a highly-sensitive non-radioactive bioluminescent assay using the PPiLight Pyrophosphate Detection Kit (Lonza) as reported previously[59]. The viral dUTPase hydrolyzes dUTP to dUMP and PPi. The detection reagent catalyzes the conversion of AMP and the enzymatically-produced PPi to ATP. Then, the assay uses luciferase, which produces light from the newly formed ATP and luciferin. This method was used to assay vdUTPase activity in 0.3 μM purified 6xHis-SUMO-vdUTPase or 6xHis-SUMO-vdUTPaseD97A in the presence or absence of 0.3 μM purified SE-piUL49 or SE-piUL49R83L. In the white half-wall of 96-well plates (Corning), each recombinant protein was mixed with 10 μl of 5× reaction buffer (250 mM Tris-HCl [pH 8.0], 10 mM β-mercaptoethanol, 5 mM MgCl$_2$, 0.5% bovine serum albumin, and 10 mM p-nitrophenylphosphate), 12.5 μl of PPiLight converting reagent (Lonza) and 12.5 μl of PPiLight detection reagent (Lonza). After incubation at 4 °C for 30 min, the vdUTPase reactions were performed by adding 10 μl of high purity dUTP (GeneAct). Finally, luminescence was measured at 37 °C for 30 min using the Multimode plate reader Enspire (PerkinElmer). In all experiments, the produced relative luminescence units (RLUs) were directly proportional to concentrations of the dUTPase-produced PPi (Supplementary Fig. 17b). RLUs were normalized by subtracting the value of vdUTPaseD97A from that of vdUTPase at each timepoint. Initial velocities ($V_0$) of the PPi synthesis by vdUTPase were determined from the maximum slope of four timepoints of the progress curve, and the maximum reaction velocity ($V_{max}$) and $K_m$ values were calculated using GraphPad Prism 6 (GraphPad Software Inc). In other studies (Supplementary Fig. 17c–e), purified 6xHis-SUMO-vdUTPase, 6xHis-SUMO-vdUTPaseD97A or SE-piUL49 in the white half-wall of 96-well plates (Corning) were mixed with 10 μl of 5× reaction buffer (250 mM Tris-HCl [pH 8.0], 10 mM β-mercaptoethanol, 5 mM MgCl$_2$, 0.5% bovine serum albumin, and 10 mM p-nitrophenylphosphate) in the absence or presence of 0.92, 2.74, or 8.23 μM dUTP (GeneAct). The reaction mixtures were incubated at 37 °C for 6 h. Then, samples were mixed with 25 μl of PPiLight converting reagent (Lonza) and 25 μl of PPiLight detection reagent (Lonza) in the absence or presence of 0.92, 2.74, or 8.23 μM PPi (Wako). After incubation at 37 °C for 30 min, luminescence was measured using the Multimode plate reader Enspire (PerkinElmer), and RLUs were normalized by subtracting the value of each reaction from that of vdUTPaseD97A.

**Phos-tag SDS-PAGE analysis**. Phos tag (phosphate affinity) SDS-PAGE analyses were performed according to the manufacturer's instructions. Briefly, Vero cells were mock infected or infected with HSV-1(F) or iUL49ΔM at an MOI of 510 for 12 h and then harvested, solubilized in SDS sample buffer (62.5 mM Tris-HCl [pH 6.5], 20% glycerol, 2% SDS, 5% 2-mercaptoethanol), and analyzed by electrophoresis in a denaturing gel containing 90 μM MnCl$_2$ and 45 μM Phos-tag acrylamide (Wako). After electrophoresis, gels were soaked in standard transfer buffer (200 mM Tris, 384 mM glycine, 20% methanol) with 1 mM EDTA for 10 min to remove Mn$^{2+}$. The separated proteins in the denaturing gels were then transferred to nitrocellulose membranes and analyzed by immunoblotting as described above.

**ELISA**. Mouse-IL-1β in the culture supernatants were quantitated by ELISA. Nunc-Immuno plates (Thermo Fisher Scientific) coated with the monoclonal anti-mouse/rat-IL-1β(eBioscience) were blocked with 2% FCS in PBS and cell culture supernatant were added to the plates. The indicated biotin-conjugated detection rabbit polyclonal biotin-anti-mouse-IL-1β (eBioscience), avidin-HRP (eBioscience), and TMB solution (eBioscience) were further added to the plates and the captured cytokines were detected by a Perkin Elmer EnSpire multimode plate reader.

**Identification of proteins that interact with piUL49**. HEK293 cells were mock-transfected or transfected with pFlag-iUL49. Flag-piUL49 was lysed with lysis buffer (10 mM 4-(2-hydroxyethyl)-1-piperazineethanesulfonic acid [HEPES] [pH 7.5], 150 mM NaCl, 50 mM NaF, 1 mM Na$_3$VO$_4$, 5 μg/mL leupeptin, 5 μg/mL aprotinin, 3 μg/mL pepstatin A, 1 mM phenylmethylsulfonyl fluoride [PMSF], and 1 mg/mL digitonin) and cleared by centrifugation. The cleared lysate was immunoprecipitated with Dynabeads Protein G (Thermo Fisher Scientific) reacted with anti-Flag M2 antibody (Sigma). The beads were then washed three times with wash buffer (10 mM HEPES [pH 7.5], 150 mM NaCl, 0.1% Triton X-100) and eluted with Flag elution buffer (0.5 mg/mL Flag peptide, 10 mM HEPES [pH 7.5], 150 mM NaCl, and 0.05% Triton X-100). Eluted proteins were precipitated with the TCA/acetone precipitation method[60]. Precipitated proteins were redissolved in guanidine hydrochloride and reduced with TCEP, alkylated with iodoacetamide, followed by digestion with lysyl endopeptidase (Wako) and trypsin (Thermo Fisher Scientific). The digested peptide mixture was applied to a Mightysil-PR-18 (Kanto Chemical) frit-less column (45 3 0.150 mm ID) and separated using a 0–40% gradient of acetonitrile containing 0.1% formic acid for 80 min at a flow rate of 100 nL/min. Eluted peptides were sprayed directly into a mass spectrometer (Triple TOF 5600 + ; AB Sciex). MS and MS/MS spectra were obtained using the information-dependent mode. Up to 25 precursor ions above an intensity threshold of 50 counts/s were selected for MS/MS analyses from each survey scan. All MS/MS spectra were searched against protein sequences of the RefSeq (NCBI) human protein plus HSV-1 protein database using the Protein Pilot software package (AB Sciex) and its decoy sequences. Then the peptides were identified with high confidence (false discovery rate [FDR] of <1%).

**Assay for cell viability**. The viability of sh-Luc-HEp-2 and sh-hDUT-HEp-2 cells was assayed using a cell counting kit-8 (Dojindo) according to the manufacturer's instructions.

**Statistical analysis**. Differences in the vdUTPase activity, IL1β amounts, $V_{max}$ values, $K_{cat}$ values, $K_m$ values, plaque sizes, luciferase signaling, or dUTPase activity in the cell cultures or mice were statistically analyzed using ANOVA and Tukey's test. Differences in viral yields in the brains of mice, viral yields from cell cultures or vdUTPase amounts were statistically analyzed using ANOVA and Tukey's test or a two-tailed Student's $t$ test. Differences in viral yields from the eyes, TG or brains of mice, protein concentrations, peptide concentrations, numbers of PLA puncta per cell, dUTPase amounts, or relative cell viability were statistically analyzed using a two-tailed Student's $t$ test or Welch's $t$-test. Differences in the mortality of infected mice were statistically analyzed by the Log-rank test. A $P$ value < 0.05 was considered statistically significant. Student's or Welch's $t$-tests were performed in Microsoft Excel (Version 15. 23). Multiple comparison and Log-rank tests were performed in Microsoft Excel for MAC or GraphPad Prism 6 for MAC OS X (GraphPad Software, San Diego, CA). No methods were used to determine whether the data meet assumptions of the statistical approach.

**Reporting summary**. Further information on research design is available in the Nature Research Reporting Summary linked to this article.

## Data availability

The MS data have been deposited with the Japan Proteome Standard Repository/Database (jSPOT) under an accession key 1618 [https://repository.jpostdb.org/preview/7697567455ed8d6772bd03] (JPST000586) and are provided in Supplementary Data 1. All data that support the findings of this study are either included in this published article and its Supplementary Information or available from the corresponding author upon request. Source data are provided with this paper.

## Code availability

The algorithms for search procedure for conservation can be found at Supplementary Software 1.

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

## Acknowledgements

We thank Tohru Ikegami, Risa Abe, Keiko Sato, and Yoshie Asakura for their excellent technical assistance. We are grateful to Tadashi Suzuki for helpful discussions with regard to enzyme analyses and Hiroaki Takeuchi for providing valuable reagents. This study was supported by Grants for Scientific Research and Grant-in-Aid for Scientific Research (S) (20H05692) from the Japan Society for the Promotion of Science (JSPS), Grants for Scientific Research on Innovative Areas from the Ministry of Education, Culture, Science, Sports and Technology of Japan (16H06433, 16H06429, 16H06430, 16K21723, 17H05610, 18H04968, 20H04853, and 24115007), contract research funds from the Program of Japan Initiative for Global Research Network on Infectious Diseases (J-GRID) (JP18fm0108006) and the Research Program on Emerging and re-emerging Infectious Diseases (19fk018105h0001, 20wm0125002h0001, 20wm0225017s, and 20wm0225009h) from the Japan Agency for Medical Research and Development (AMED), the grant for International Joint Research Project of the Institute of Medical Science, the University of Tokyo, grants from the Takeda Science Foundation, the Naito Foundation, and the Ichiro Kanehara Foundation, the Kanae Foundation for the Promotion of Medical Science, the Waksman Foundation of Japan, and the GSK Japan Research Grant 2018. The computational resource was provided by the Super Computer System, Human Genome Center, Institute of Medical Science, and The University of Tokyo.

## Author contributions

A.K., S.A., T.N., and Y.K. conceived this study; A.K., S.A., K.T., MW., N.K., Y.M., J.A., Shuichi K., R.S., and T.H. conducted the experiments; Shuichi K. and H.K. performed bioinformatic analysis; A.K., S.A., Shinobu K., T.N., and Y.K. discussed the results; and A.K. and Y.K. wrote the manuscript.

## Competing interests

The authors declare no competing interests.
