## [Peer Review File · Nature Communications]

Editorial Note: This manuscript has been previously reviewed at another journal that is not operating a transparent peer review scheme. This document only contains reviewer comments and rebuttal letters for versions considered at Nature Communications .

Reviewers' Comments:

Reviewer #6:

Remarks to the Author:

The manuscript "Discovery of a Cryptic Orphan Herpes Simplex Virus 1 Gene Encoding a Neurovirulence Factor by Chemical Proteomics " describes a very nice application of Azidohomoalanine (AHA) labeling to identify HSV1 ORFs that are being translated and could code for novel viral proteins. The work shows that one of the ORFs, iUL49, encodes a neural virulence factor by showing that intracranial infection of piUL49ΔM reduces viral titer compared to wt in the brain and augments LD50 values, although the effect on the viral titer is much smaller than the effect of VP22ΔM virus for example. The authors then go on to show that piUL49 interacts with vdUTPase in Vero cells, and increases its enzymatic activity, which is critical for viral replication under conditions when host dUTPase activity is low such as in the brain. The authors thus suggest that this the mechanism by which piUL49 affects viral replication in the brain.

The authors have carefully addressed a number of issues that were brought up, in particular they have now described the identification of AHA labeled proteins by Mass Spectrometry in more detail, re-analyzed the data in a more appropriate way and provided raw data as well as representative spectra. They have also corrected the statistical approaches.

Before publication, the authors need to address the two comments below.

1. What was the intensity threshold used for ms2 spectra ? Some of the representative spectra in Supplementary Fig 2 (especially Supplementary Figure 2k, 2o, 2s, 2q and 2f) are very much in the noise. This is not very confidence inspiring. If the authors have better spectra for the affected cryptic ORFs they should replace the spectra. If other spectra are also in the same intensity range, it is questionable that those ORFs were really identified. The intensity units are missing from the figure as well.

2. In Supplementary table 2, "mass deviation" should indicate the unit. I believe it is given in ppm ? A mass deviation of -1741.391 (ppm ?) is given for the peptide "PGLHAPVYQSAVSGPAGR " used for identification of iUs. If this is not a comma mistake, the peptide needs to be removed as its mass deviation is much too large to be a true hit.

Point-by-point response to the reviewer's comments

Please find our point-by-point responses to your comments below.

Reviewer 6

General Comments: Explanation and evaluation of the manuscript. Before publication, the authors need to address the two comments below.

Response: Please see the responses to the comments below.

Comment 1: What was the intensity threshold used for ms2 spectra ? Some of the representative spectra in Supplementary Fig 2 (especially Supplementary Figure 2k, 2o, 2s, 2q and 2f) are very much in the noise. This is not very confidence inspiring. If the authors have better spectra for the affected cryptic ORFs they should replace the spectra. If other spectra are also in the same intensity range, it is questionable that those ORFs were really identified. The intensity units are missing from the figure as well.

Response: We did not use the intensity threshold for the MS/MS spectra. In the original manuscript, we showed the spectra based on profile spectrum data, which were very much in the noise as the reviewer suggested. Peptide sequences are usually determined based on centroid spectrum data (Page 18, line 391 in the revised manuscript) and the spectra based on these data have been displayed in earlier publications (Nat. Methods 16: 894-901, 2019; Nature 558: 435-439, 2018; Sci. Rep. 6: 37189, 2016; Nat. Protoc. 3: 1630-1638, 2008). Therefore, in the revised manuscript, we replaced the spectra with those based on centroid spectrum data, which are much better than the spectra in the original manuscript (Supplementary Fig. 2). We also added the unit for Y-axis (cps: count per second) in Supplementary Fig. 2 in the revised manuscript.

Comment 2: In Supplementary table 2, “mass deviation” should indicate the unit. I believe it is given in ppm ? A mass deviation of -1741.391 (ppm ?) is given for the peptide “PGLHAPVYQSAVSGPAGR “ used for identification of iUs. If this is not a comma mistake, the peptide needs to be removed as its mass deviation is much too large to be a true hit.

Response: The value of mass deviation given for this peptide was much too large because multiple peptides were simultaneously analyzed in the MS/MS analysis. Therefore, as the reviewer suggested, we have removed the information on the peptide

in the revised manuscript (Supplementary Table 2). We also added the unit for mass deviation (ppm) in Supplementary Table 2 in the revised manuscript.

Reviewers' Comments:

Reviewer #6:

Remarks to the Author:

The authors have satisfactorily answered the questions of the reviewer.

The replaced spectra are now much more confidence inspiring than the ones shown in the original manuscript and support the authors conclusions.

Point-by-point response to the reviewer's comments

Please find our point-by-point responses to your comments below.

Reviewer 6

Comments: The authors have satisfactorily answered the questions of the reviewer. The replaced spectra are now much more confidence inspiring than the ones shown in the original manuscript and support the authors conclusions.

Response: None.